# Automatic Detection of the Circulating Cell-Free Methylated DNA Pattern of *GCM2*, *ITPRIPL1* and *CCDC181* for Detection of Early Breast Cancer and Surgical Treatment Response

**DOI:** 10.3390/cancers13061375

**Published:** 2021-03-18

**Authors:** Sheng-Chao Wang, Li-Min Liao, Muhamad Ansar, Shih-Yun Lin, Wei-Wen Hsu, Chih-Ming Su, Yu-Mei Chung, Cai-Cing Liu, Chin-Sheng Hung, Ruo-Kai Lin

**Affiliations:** 1Ph.D. Program in Drug Discovery and Development Industry, College of Pharmacy, Taipei Medical University, No. 250, Wuxing Street, Taipei 110, Taiwan; d343107002@tmu.edu.tw; 2Division of General Surgery, Department of Surgery, Taipei Medical University Shuang Ho Hospital, No.291, Zhongzheng Rd., Zhonghe District, New Taipei City 23561, Taiwan; b9002008@cgu.edu.tw (L.-M.L.); 08261@shh.org.tw (C.-M.S.); 3Ph.D. Program in the Clinical Drug Development of Herbal Medicine, Taipei Medical University, 250 Wu-Hsing Street, Taipei 110, Taiwan; d339107004@tmu.edu.tw; 4Graduate Institute of Pharmacognosy, Taipei Medical University, 250 Wu-Hsing Street, Taipei 110, Taiwan; m303108006@tmu.edu.tw; 5Department of Statistics, College of Arts and Sciences, Kansas State University, 101 Dickens Hall, 1116 Mid-Campus Drive N, Manhattan, KS 66506-0802, USA; wwhsu@ksu.edu; 6Department of Surgery, School of Medicine, College of Medicine, Taipei Medical University, No. 250, Wuxing Street, Taipei 110, Taiwan; 7Master Program for Clinical Pharmacogenomics and Pharmacoproteomics, Taipei Medical University, 250 Wu-Hsing Street, Taipei 110, Taiwan; m338104001@tmu.edu.tw; 8School of Medical Laboratory Science and Biotechnology, College of Medical Science and Technology, Taipei Medical University, 250 Wu-Hsing Street, Taipei 110, Taiwan; b614102044@tmu.edu.tw; 9Clinical trial center, Taipei Medical University Hospital, 252 Wu-Hsing Street, Taipei 110, Taiwan

**Keywords:** breast cancer, DNA methylation, early detection, circulating cell-free DNA, automatic detection, surgical treatment response, Recursive Partitioning and Regression Trees, *CCDC181*, GCM2 and ITPRIPL1

## Abstract

**Simple Summary:**

Breast cancer is a multifactorial disease that arises from the cumulative accumulation of acquired genetic as well as epigenetic alterations. Epigenetic alteration constitutes a molecular signature that can serve as a promising biomarker for early detection. In this study, we carry out automatic detection of circulating cell-free methylated DNA, including *GCM2*, *ITPRIPL1* and *CCDC181*, and find a pattern that can detect early breast cancer more accurately compared with currently used biomarkers. The pattern is also useful for the detection of the surgical responses of breast cancer patients. Circulating methylated *CCDC181*, *GCM2* and *ITPRIPL1* analysis could be combined with ultrasound to facilitate the early detection of breast cancer.

**Abstract:**

The early detection of cancer can reduce cancer-related mortality. There is no clinically useful noninvasive biomarker for early detection of breast cancer. The aim of this study was to develop accurate and precise early detection biomarkers and a dynamic monitoring system following treatment. We analyzed a genome-wide methylation array in Taiwanese and The Cancer Genome Atlas (TCGA) breast cancer (BC) patients. Most breast cancer-specific circulating methylated *CCDC181*, *GCM2* and *ITPRIPL1* biomarkers were found in the plasma. An automatic analysis process of methylated ccfDNA was established. A combined analysis of *CCDC181*, *GCM2* and *ITPRIPL1* (CGIm) was performed in R using Recursive Partitioning and Regression Trees to establish a new prediction model. Combined analysis of *CCDC181*, *GCM2* and *ITPRIPL1* (CGIm) was found to have a sensitivity level of 97% and an area under the curve (AUC) of 0.955 in the training set, and a sensitivity level of 100% and an AUC of 0.961 in the test set. The circulating methylated *CCDC181*, *GCM2* and *ITPRIPL1* was also significantly decreased after surgery (all *p* < 0.001). The aberrant methylation patterns of the *CCDC181*, *GCM2* and *ITPRIPL1* genes means that they are potential biomarkers for the detection of early BC and can be combined with breast imaging data to achieve higher accuracy, sensitivity and specificity, facilitating breast cancer detection. They may also be applied to monitor the surgical treatment response.

## 1. Introduction

Worldwide, breast cancer (BC) is the most commonly diagnosed cancer in women [1,2]. Among 25–64-year-old women, breast cancer is the leading cause of cancer death in Taiwan [2]. In the United States and Taiwan, breast cancer has been the most common cancer for the past 9 years and is now also the third leading cause of cancer death in females and males [1,2,3]. Breast cancer is curable in ~90% of cases if it is detected at an early stage. Early detection of breast cancer through screening programs reduces the incidence and mortality rates of this disease. Investigators have studied many breast diagnostic approaches, including mammography and ultrasound [4]. Evidence shows that national mammography screening programs have sufficiently reduced BC-related mortality. However, both false-positive and false-negative BC diagnoses, excessive biopsies, irradiation linked to mammography application as well as suboptimal mammography-based screening in some populations, such as young females with highly dense breast tissue and Asian women, are limitations of mammography-based screening [5,6]. The sensitivity of mammography is related to age, ethnicity, personal history, the radiologist’s experience and technique quality [6]. Breast ultrasonography is a cost-effective and widely available screening tool, in which tumors are detected by bouncing acoustic waves off breast tissue. However, breast ultrasonography fails to detect many tumors because the acoustic properties of healthy and cancerous tissues are very similar. Moreover, it requires experienced radiologists, which significantly affects the sensitivity and specificity of the test [4]. There is no clinically useful, noninvasive biomarker for the early prediction of breast cancer. Cancer antigen 15-3 (CA-153) is widely used for the detection of recurrent breast cancer in current clinical practice, but not in the early stage of breast cancer. Even in some patients with metastatic breast cancer, the sensitivity of CA-153 is only 60 to 70% [7]. Simultaneous use of the serum markers CA-153 and carcinoembryonic antigen (CEA) leads to the early diagnosis of metastasis in only 60–80% of patients with breast cancer [8]. Even a combined analysis of a serum panel of potential breast cancer markers, consisting of osteopontin, haptoglobin, haptoglobin, CA153, CEA, CA-125, prolactin, CA19-9, α-fetoprotein, leptin and migration inhibitory factor, is unable to predict the presence of early-stage breast cancer [9].

Breast cancer is a multifactorial disease that arises from the cumulative accumulation of acquired genetic as well as epigenetic alterations in a number of oncogenes, tumor suppressor genes (TSGs), mismatch repair genes and cell cycle genes during tumorigenesis [10,11,12,13]. Among the epigenetic alterations that occur, gene silencing by aberrant DNA methylation of promoter regions remains the most dominant phenomenon during tumorigenesis [14]. Regardless of the biological consequences of methylation-induced silencing of tumor suppressor genes, this epigenetic alteration constitutes a molecular signature that can serve as a promising biomarker to detect early-stage disease, progression and ultimately recurrence and metastasis [13,14,15,16]. Therefore, aberrant DNA methylation could have potential to be used as an accurate, precise and dynamic monitoring system for early prediction, complete treatment response and recurrence prediction in current clinical practice.

Circulating cell-free DNA (ccfDNA) released from apoptotic or necrotic tumor cells contain genetic defects and molecular features that are identical to the tumor tissue and/or cells from which they originate [17]. Circulating cell-free methylated DNA (cfmeDNA) assays could serve as outcome predictors in chemotherapy and multikinase inhibitor-treated metastatic breast cancer patients [18]. Circulating cell-free DNA levels been shown to have a greater dynamic range and better correlation with changes in tumor burden than CA-153 or circulating tumor cells [19]. Previous studies have described a reduction of aberrant DNA methylation in several genes following surgery in breast cancer patients, such as *aldo-keto reductase family 1*, *member B1*, *Rho guanine nucleotide exchange factor 7*, *collagen type VI alpha 2 chain*, *glutathione peroxidase 7*, *transmembrane 6 superfamily member 1*, *transmembrane protein with EGF like and two follistatin like domains 2*, *homeobox B4*, *Ras protein specific guanine nucleotide releasing factor 2*, *Ras association domain family member 1* (*RASSF1*), *Histone H3*, *retinoic acid receptor beta*, *mutS homolog 2* and *O-6-methylguanine-DNA methyltransferase* [20,21]. These results suggest that aberrant DNA methylation could be a marker for early prediction and monitoring of the treatment response. Other applications also include, but are not limited to, cancer screening, early diagnosis, prognostic assessment, evaluation and management of preoperative systemic or local therapies, postsurgical detection of minimal residual disease and early detection of cancer relapse, and assessment of the emergence of drug resistance, survival and monitoring tumor burden [22,23,24,25]. These findings encouraged us to discover noninvasive cfmeDNA biomarkers for breast cancer. Genome-wide methylation array analysis of Taiwanese tissues combined with The Cancer Genome Atlas (TCGA) data were used to select breast cancer-specific hypermethylated genes and biomarkers. The results were validated in paired Taiwanese breast cancer tissues and plasma samples to establish a blood-based prediction model using manual and automatic techniques. The automatic cfmeDNA analysis was further applied to monitor the surgery treatment response. Our aims were to develop accurate and precise early detection biomarkers and a dynamic monitoring system for residual tumors following surgery treatment, with the exception of primary breast tumors.

## 2. Materials and Methods

### 2.1. Patients and Sample Preparation

A total of 555 samples, including 109 human breast tumor tissues samples, 109 adjacent-normal breast tissue samples and 446 plasma samples, were obtained from Taipei Medical University (TMU) Hospital, Shuang Ho Hospital and the TMU Joint Biobank from 2013 to 2020 (Figure 1). Before clinical data and sample collection, written informed consent was obtained from all patients. Healthy women aged 40–80 years, with Breast Imaging Reporting and Data System (BI-RADS) classifications of 1 to 3 for a follow-up period of at least 6 months using ultrasound or diagnosed as having benign tumors by core needle biopsy, were identified. All breast cancer patients were diagnosed with tissue proof. Lesion samples obtained by core needle biopsy were evaluated by pathologists and classified into histological categories. Patients undergoing an emergent operative procedure were excluded from this study. Sections of cancerous tissue and corresponding noncancerous tissues were reviewed by a senior pathologist. Clinical data on age, gender, tumor type, TNM tumor stage, BI-RADS, menopause state, estrogen receptor (ER), progesterone receptor (PR), human epidermal growth factor receptor 2 (HER2), Ki-67, tumor markers, CA153 and CEA, which were prospectively collected, were obtained from Taipei Medical University (TMU) Hospital, Shuang Ho Hospital and TMU Joint Biobank. Following surgery, patients were monitored every 3 months for the first 2 years and semiannually thereafter. For the plasma sample analysis, (1) the first patient pool (0.2 mL plasma sample) was obtained from the TMU biobank, used for first biomarkers screening; (2) the patient pool (0.5 mL plasma sample) was recruited from Taipei Medical University (TMU) Hospital, Shuang Ho Hospital, which was used for the second biomarkers screening; and (3) to enhance the sensitivity and stability of the analysis, we performed the assay in a third patient pool (1.6 mL plasma sample). The third patient pool was also recruited from Taipei Medical University (TMU) Hospital and Shuang Ho Hospital. The clinical parameters of the three patient pools are shown in Appendix A. Blood samples were collected using an ETDA-K2 tube and PAXgene Blood ccfDNA (circulating cell-free DNA) tube (Qiagen, Hilden, Germany, 768165) designed specifically for in vitro diagnostic ccfDNA testing. The samples collected using ETDA-K_2_ tube (BD, Plymouth, UK, 367525) were immediately centrifuged at 2000× *g* for 10 min at 4 °C. Within 2 h, the supernatant from each sample was transferred to a new centrifuge tube and centrifuged at 6000× *g* for 30 min at 4 °C and subsequently stored at −80 °C. Samples collected using the PAXgene Blood ccfDNA tube were kept at room temperature (15–25 °C) until use within 3 days, and subsequently centrifuged at 2000× *g* for 10 min at 4 °C followed by 6000× *g* for 30 min at 4 °C for plasma separation. The plasma of each sample was split into 1.6 mL aliquots and immediately frozen at −80 °C until further use.

### 2.2. The Cancer Genome Atlas Portal

The data of the Western cohort are based on data generated by The Cancer Genome Atlas (TCGA) Research Network from Genomic Data Commons (GDC) data portal with release version 1.0. (http://cancergenome.nih.gov/ accessed on 6 June 2016) and were downloaded between 2015 and 2017. The Cancer Genome Atlas (TCGA) is a collaboration between the National Cancer Institute (NCI) and the National Human Genome Research Institute (NHGRI) that has generated comprehensive, multidimensional maps of the key genomic changes in 33 types of cancer. The TCGA dataset, comprising more than two petabytes of genomic data, is now accessible to the cancer research community to improve the prevention, diagnosis and treatment of cancer. There are 87 paired breast cancer and noncancer tissues: 88 tumors of triple-negative breast cancer (TNBC) and 7 pairs of TNBC, 1098 breast cancer from the TCGA portal.

### 2.3. Genomic DNA Extraction

Genomic DNA from matched pairs of primary tumors and adjacent breast tissues from the same patient was extracted using the QIAamp DNA Mini Kit (Qiagen, Bonn, Germany, Cat. No. 51306) according to manufacturer’s instruction. After DNA quantification, the purity was verified by measuring the A260/A280 ratio (range 1.8 to 2.0) using a NanoDrop ND-1000 spectrophotometer (NanoDrop Technologies Inc., Wilmington, DE, USA).

### 2.4. Manual Circulating Cell-Free DNA Extraction

Circulating cell-free DNA (cfDNA) from plasma samples was extracted using the MagMAX Cell-Free DNA Isolation Kit according to the manufacturer’s recommended protocol (Thermo Fisher Scientific, Austin, TX, USA) [26,27]. The ccfDNA samples had clear fragment size peaks between 140 and 200 bp [28]. The MagMAX Cell-free DNA Isolation kit provided the highest yield and low molecular weight fractions [28,29]. The plasma was isolated immediately from 10 mL of peripheral blood within 2 h. After DNA quantification, the purity was verified by measuring the A260/A280 ratio (range 1.8 to 2.0) using a NanoDrop ND-1000 spectrophotometer (NanoDrop Technologies, Inc., Wilmington, DE, USA).

### 2.5. Automatic Circulating Cell-Free DNA Extraction and Bisulfite Conversion by KingFisher™ Duo Prime

An automated process for ccfDNA extraction and bisulfite conversion on the KingFisher™ Duo Prime purification system (ThermoFisher Scientific, Singapore, Singapore) was applied according to the manufacturer’s instructions. This process fully automates magnetic bead-based DNA extraction of up to six samples simultaneously. The workflow was adapted as described in the instruction manual supplied with the MagMAX™ cell-free DNA isolation kit (ThermoFisher Scientific, Austin, TX, USA, A29319). The ccfDNA was extracted from 1.6 mL of plasma and eluted in 60 µL of molecular biology-grade water (Corning, NY, USA, 46-000-CM). The bisulfite conversion cleanup was also performed on this machine for the semiautomatic assay. The automated protocol for the bisulfite conversion cleanup was developed with the instruction manual supplied with the EZ-96 DNA Methylation-Lightning™ MagPrep Kit (Zymo Research, Irvine, CA, USA, D5046). The extracted ccfDNA was incubated with sodium bisulfate (6 M) and hydroquinone (10 mM) in a 60 °C incubator for 30 min following by the automated process. We used 60 µL of ccfDNA for bisulfite conversion, and the bisulfite-converted ccfDNA was eluted in 100 µL of molecular biology-grade water. The automated sample process was performed using a 24 deep-well plate (ThermoFisher Scientific, Vantaa, Finland, 95040470). The eluted bisulfite-converted ccfDNA was immediately used for methylation-specific real-time PCR.

### 2.6. Automatic Circulating Cell-Free DNA Extraction Using LabTurbo 24C

The automated ccfDNA extraction process was performed using the LabTurbo 24 Compact System (Taigen Bioscience Co., Taipei, Taiwan) according to the manufacturer’s instructions. The workflow followed the instruction manual supplied with the Labturbo Circulating DNA mini kit (Cat No. AIOLCD1600, Taigen Bioscience Co., Taipei, Taiwan), with full automation of vacuum-based DNA extraction of up to 24 samples simultaneously. The ccfDNA was extracted from 1.6 mL of plasma and eluted in 60 µL of molecular biology-grade water (46-000-CM, Corning, NY, USA).

### 2.7. MethylationEPIC BeadChip Array for Genome-Wide Methylation Analysis

The MethylationEPIC BeadChip (EPIC) array covers 850,000 CpG sites, including >90% of the CpGs and 99% Refseq genes from HM450 and an additional 413,743 CpGs [30]. The EPIC array has been validated in comparison to the 450K platform for blood samples [30]. The genome-wide methylation analysis was performed using the Infinium^®^ MethylationEPIC BeadChip array (Illumina, San Diego, CA, USA). Bisulfite conversion was performed for 500 ng of DNA using the EpiTect Fast DNA Bisulfite Kit (QIAGEN, Bonn, Germany, Cat. No. 59826) according to the manufacturer’s instructions. Methylation scores for each CpG site are represented as “beta” values ranging from 0 (unmethylated) to 1 (fully methylated) by determining the ratios of the methylated signal intensities to the sums of the methylated and unmethylated signal outputs. Infinium MethylationEPIC BeadChip data were analyzed using GenomeStudio Methylation Module version 2011.1. The Infinium MethylationEPIC BeadChip employs both Infinium I and Infinium II assays. The Infinium I assay design employs 2 bead types per CpG locus, 1 each for the methylated and unmethylated states. The Infinium II design uses 1 bead type, with the methylated state determined at the single base extension step after hybridization (right panel). A differentially methylated CpG heatmap of the target genes was visualized by a heatmap using heatmapper software. A gradient-scale heatmap (100 color categories) was used to visualize the DNA methylation level from low to high (yellow to blue) [31].

### 2.8. Probe-Based Quantitative Methylation-Specific PCR (qMSP)

After bisulfite conversion of DNA, which was done according to the manufacturer’s recommended protocol, the DNA methylation levels of the candidate genes *coiled-coil domain containing 181* (*CCDC181*), *glial cells missing transcription factor 2* (*GCM2*), *ITPRIP like 1* (*ITPRIPL1*), *ectonucleotide pyrophosphatase/phosphodiesterase 2* (*ENPP2*), *secretoglobin family 1B member 2*, *pseudogene* (*LOC643719*), *zinc finger protein 177* (*ZNF177*), *adenylate cyclase 4* (*ADCY4*) and *Ras association domain family member 1* (*RASSF1*) were measured using TaqMan quantitative methylation-specific PCR (qMSP) with a LightCycler 96 (Roche Applied Science, Penzberg, Germany). qMSP was performed using the SensiFAST™ Probe No-ROX Kit (Bioline, London, UK, Cat. No. BIO-86020) with specific primers and methyl-TaqMan probes of candidate genes. Normalized DNA methylation values, which were calibrated to the control group, were obtained using LightCycler Relative Quantification software (Version 1.5, Roche Applied Science). The *beta-actin* (*ACTB*) gene was used as methylation-independent DNA control. The primers/probes for the *ACTB* gene were designed without the CpG site (as a control for input DNA) [32,33]. The primers/probes for candidate genes were designed on their methylated promotor regions, especially on the identified differential regions between normal and tumor tissues. A total of 6 CpGs were designed on the *CCDC181* gene, 7 CpGs were designed on the *GCM2* gene and 5 CpGs were designed on the *ITPRIPL1* gene. According to the sequencing results, only when all CpG sites are methylated can a successful PCR reaction occur. The target genes were considered hypermethylated when the methylation level relative to that of the *ACTB* gene was at least 2-fold higher in the breast tumor compared with the paired normal breast tissue sample. The specificity of the candidate gene methylation end products was confirmed by bisulfite sequencing (Appendix A). The primers and probes used for qMSP are listed in Appendix A.

### 2.9. Breast Cancer Detection Model Construction

Further analysis was performed by Recursive Partitioning and Regression Trees (RPART) using data obtained from the qMSP assays [34]. The original data were processed according to the following criteria. If the PCR results for the target gene show a positive signal, they are defined as 1; if they show a negative signal, they are defined as 0. To determine the methylation value of each gene, the results were summed from the times for positive target gene reactions and then divided by the number of PCR technical replicates. If all the triplicates of PCRs were positive, the sum of the specific gene would be 1. The CGI total methylation value of each sample was then summed from the results of each gene. If all three genes showed positive PCRs for all three replicates, the sum of the CGI results was 3. The CGI methylation values were calculated using the following formula:

(1)$$\begin{array}{ll} CGIm\;value= & \frac{sum(times\;for\;positve\;CCDC181\;PCR\;reaction)}{\mathrm{times}\;\mathrm{for}\;\mathrm{PCR}\;\mathrm{technical}\;\mathrm{replicates}}\\ & \;+\frac{sum(times\;for\;positve\;GCM2\;PCR\;reaction)}{\mathrm{times}\;\mathrm{for}\;\mathrm{PCR}\;\mathrm{technical}\;\mathrm{replicates}}\\ & \;+\frac{sum(times\;for\;positve\;ITPRIPL1\;PCR\;reaction)}{\mathrm{times}\;\mathrm{for}\;\mathrm{PCR}\;\mathrm{technical}\;\mathrm{replicates}} \end{array}$$

According to the principle of simple random sampling, the 191 cases were divided into two sets: 153 cases (80%) in the training set and 38 cases (20%) in the test set. The dataset was subdivided into 10 subsets, and the process was repeated 10 times with 8 parts for model training and the last 2 parts as the test dataset. The training set was used to build a decision tree model, which was then verified by the test set. This process was repeated 10 times, building 10 separate test trees. The 10 results could then be averaged to produce a single evaluation of modeling effectiveness.

To measure the best split of the methylation level between patient and health samples, the classification was performed in R (version 4.0.0; The R Foundation for Statistical Computing; http://www.r-project.org/foundation accessed on 6 June 2016) using the RPART package (version 4.1–15). Decision trees from the RPART model were plotted using the partykit package (version 1.2-7) in R. The performance of this decision tree was plotted using the pROC package (version 1.16.2) in R.

### 2.10. Statistical Analysis

The Pearson’s chi-squared test, Mann–Whitney *U* test, Wilcoxon test and Spearman’s rank correlation analyses were performed using SPSS (IBM, Armonk, NY, USA). The Pearson’s chi-squared test was used to compare breast cancer patients in terms of candidate gene methylation, RNA expression and other clinical data, including age, gender, tumor type, TNM tumor stage, race, menopause state, BI-RADS, CA-153, CEA, ER, PR, HER2 and ki-67 status. The paired-sample Wilcoxon test and *t*-test was used to compare differences in DNA methylation between tumors and matched adjacent normal tissues, different cancer types, as well as in candidate ccfDNA methylation between surgery treatment in breast cancer patients. The Spearman’s rank correlation was adopted to analyze the methylation levels of the tumor and plasma samples.

A univariate logistic regression analysis was conducted to assess the classification accuracy when only ultrasound screening was used for the detection of breast cancer [35]. To assess multiple biomarkers, Kang’s nonparametric stepwise classification method was employed to evaluate the accuracy in identifying breast cancer patients when the proposed gene biomarkers were used. In addition to the accuracy, other commonly used measures for evaluating the classification, such as the area under the receiver operating characteristic curve (AUC), sensitivity, specificity, false-positive rate and false-negative rate, were also reported.

## 3. Results

### 3.1. Seven Potential Candidate Genes Were Identified among the Breast Cancer Patients from Taiwan and the Western World

To identify a novel potential biomarker in breast cancer patients, we used three screening criteria: (1) hypermethylation in Taiwanese patients with breast cancer; (2) hypermethylation in Western patients with breast cancer; and (3) a methylation level in normal tissues close to 0 (Figure 1). First, the Infinium Methylation Assay was applied to identify critical CpG sites from five breast cancer and paired noncancerous breast tissues. A total of 1612 genes were found to be hypermethylated when the ΔAvg β (β value in the tumor−β value in normal tissues) was > 0.4. Second, we analyzed the TCGA Illumina Infinium HumanMethylation450 BeadChip array data for 87 paired Western breast cancer patients. A total of 2865 genes were hypermethylated when ΔAvg β (β value in the tumor−β value in normal tissues) was > 0.4. A total of 1190 genes were found to be overlapped between 1612 genes in the Taiwanese cohort and 2865 genes in the TCGA cohort. This study focused on the top 200 genes with the highest methylation levels in breast tumors from TCGA. In total, 160 highly methylated genes were found when the genes identified from the Taiwanese and TCGA cohorts were combined (Appendix A). A literature review was conducted and the following previously-reported aberrantly methylated genes were excluded: *DPF1* [36], *DNM3* [16], *TLX1* [37], *CHST11* [38], *CHST3* [39], *ALX1* [40], *HNF1B* [37], *EBF1* [41], *PITX2* [42], *ITGA5* [43], *COL11A2* [44], *CFTR* [45], *NR5A2* [46], *PRKCB* [46], *SOSTDC1* [47], *DBX1* [48], *LHX8* [49], *HCK* [50], *NXPH1* [51], *NID2* [52], *TAC1* [53], *TFAP2B* [54], *F2RL3* [55], *POU4F2* [56], *H2AFY* [57], *GALR1* [58], *FOXD3* [59], *NT5E* [60], *CLIP4* [61], *ZNF454* [62], *TIMP2* [63], *GNG4* [64], *MIR129-2* [64], *TXNRD1* [65], *CPXM1* [64], *PRDM14* [64], *HOXA4* [66], *SLITRK1* [67], *AHRR* [68], *NPTX2* [69], *KCNK9* [70], *C9orf122* [71], *CRHR2* [72], *BOLL* [73], *CCDC8* [74], *MMP9* [75] and *C12orf68* [71]. The remaining candidate genes were then studied through sequencing and a gene-specific primer/probe design to determine their methylation-specific PCR reactions. The results for seven candidate genes—*CCDC181, GCM2, ITPRIPL1, ENPP2, LOC643719, ZNF177* and *ADCY4*—were successfully validated by sequencing and stable detection in ccfDNA from the plasma of Taiwanese patients with breast cancer. The seven candidate genes, *CCDC181, GCM2, ITPRIPL1, ENPP2, LOC643719, ZNF177* and *ADCY4*, were found to be hypermethylated in both the Taiwanese and Western cohorts (Figure 1). *RASSF1* has been reported to be hypermethylated in breast cancer and showed a similar pattern to the reference control in breast cancer patients from both the Taiwan and Western cohorts [76]. Therefore, *CCDC181, GCM2, ITPRIPL1, ENPP2, LOC643719, ZNF177, ADCY4* and *RASSF1* were analyzed in the following assays. The sequencing profiles are shown in Appendix A. The functions of the *CCDC181, GCM2, ITPRIPL1, ENPP2, LOC643719, ZNF177* and *ADCY4* genes are described in Appendix A [77,78,79,80,81].

### 3.2. Methylation Pattern of Seven Candidate Genes Found in Taiwanese Patients with Breast Cancer

Aberrant methylation of *CCDC181, GCM2, ITPRIPL1, ENPP2, LOC643719, ZNF177* and *ADCY4* was found in breast cancer patients from the Taiwanese and TCGA cohorts using the methylation array. To further determine the methylation pattern of breast cancer tissues in the Taiwanese cohort, we analyzed the methylation levels of *CCDC181, GCM2, ITPRIPL1, ENPP2, LOC643719, ZNF177* and *ADCY4* in tumors and adjacent normal tissue in Taiwanese breast cancer patients. A 2-fold increase in DNA methylation was detected for *CCDC181, GCM2, ITPRIPL1, LOC643719, ZNF177, ENPP2, ADCY4* and *RASSF1* in 81.5%, 70.4%, 79.6%, 66.7%, 83.0%, 77.4%, 71.7% and 73.6% of the breast cancer samples, respectively (Table 1 and Appendix A). Hypermethylation of *CCDC181* and *GCM2* could be detected in ductal carcinoma in situ (DCIS) tumors of breast cancer patients (Table 1). Hypermethylation of *CCDC181, GCM2* and *ITPRIPL1* was found in more than 80% of patients with lower tumor histologic grades, and in 70% of early-stage breast cancer among all subtypes of breast cancer patients (Table 1). There was no significant difference between hypermethylation of *CCDC181, GCM2* and *ITPRIPL1* with expression of ER and PR. While, hypermethylation of *GCM2* and *ITPRIPL1* was significantly associated with HER2 positivity (Table 1, *p* = 0.031 and 0.011).

### 3.3. Hypermethylated Circulating CCDC181, GCM2 and ITPRIPL1 Are Noninvasive Breast-Cancer-Specific Biomarkers in TCGA and Taiwanese Breast Cancer Patients

To identify noninvasive breast cancer-specific biomarkers, the methylation levels of circulating *CCDC181, GCM2, ITPRIPL1, LOC643719, ZNF177, ENPP2, ADCY4* and *RASSF1* were determined in each 200 μL plasma sample from 22 healthy and 45 breast cancer patients. The breast cancer specificity of methylated *CCDC181, GCM2, ITPRIPL1, LOC643719* and *ZNF177* ranged from 64.3 to 100%, indicating lower false-positive signals (Table 2). Methylated *ENPP2* and *ADCY4* were found in most healthy and breast cancer patients, indicating that they are not specific biomarkers for breast cancer patients. The sensitivity and specificity levels of methylated *RASSF1* were much lower: 14.3% and 33.3%, respectively (Table 2). Therefore, we selected *CCDC181, GCM2, ITPRIPL1, LOC643719* and *ZNF177* for further investigation.

To increase the sensitivity of each gene, we performed a methylation analysis with 500 μL plasma samples from another 48 healthy and 63 breast cancer patients. The increase in sensitivity of the *CCDC181, GCM2, ITPRIPL1* and *ZNF177* methylation levels ranged from 41.7 to 75.0% (Table 2). However, the detection rate of the *LOC643719* methylation levels in patients was 16.6% (Table 2). The increase in the specificity of the *CCDC181, GCM2, ITPRIPL1, ZNF177* and *LOC643719* methylation levels also ranged from 77.8 to 100.0% (Table 2).

To further select breast cancer-specific methylated genes and exclude methylated genes by organ type, 11,939 genes with very low DNA methylation levels were found in normal breast, colon, rectal, lung, uterine, gastric, esophagus, pancreases, liver and prostate tissues by analyzing the genome-wide methylation array in TCGA. The methylation levels of *CCDC181*, *GCM2* and *ITPRIPL1* were found to be much lower in normal colon, rectal, lung, uterine, gastric, esophagus, pancreas, liver and prostate tissues (Figure 1). In comparison with normal tissues, three biomarkers, *CCDC181*, *GCM2* and *ITPRIPL1*, were found to be simultaneously hypermethylated only in breast tumors in TCGA and Taiwanese patients, but not in lung cancer, uterine cancer, ovarian cancer, gastric cancer, esophagus cancer, pancreatic cancer, liver cancer or colorectal cancer (Figure 2 and Appendix A). Hypermethylation of *CCDC181*, *GCM2* and *ITPRIPL1* was observed in different subgroups of breast cancer patients, including in individuals of different races and in various menopause states as well as with different tumor stages and histological types (Appendix A). Few studies have reported the clinical significance and application of aberrant *CCDC181*, *GCM2* and *ITPRIPL1* methylation in breast cancer [82]. Therefore, the profiles of *CCDC181*, *GCM2* and *ITPRIPL1* in breast cancer were selected for further analysis.

The heatmap revealed hypermethylation of *CCDC181*, *GCM2* and *ITPRIPL1* in tumors of breast cancer patients compared with adjacent normal tissues from the Taiwanese and TCGA cohort (Figure 3A,B, respectively). The box plot demonstrates an increase in *CCDC181*, *GCM2* and *ITPRIPL1* in tumors of Taiwanese breast cancer patients compared with the adjacent normal tissues using the paired-sample Wilcoxon test (Z score −9.02, −7.94, −8.65, Figure 3C, *p* < 0.001). Hypermethylated circulating *CCDC181*, *GCM2* and *ITPRIPL1* was significantly elevated in BC patients compared with healthy subjects according to the Mann–Whitney *U* test (*Z*-score of −5.26, −4.86 and −3.38, respectively, Figure 3D, all *p* < 0.001). Circulating methylated *CCDC181*, *GCM2* and *ITPRIPL1* was detected in 57.1–84.6% of ductal carcinoma in situ (DCIS) tumors from breast cancer patients (Table 3). *ITPRIPL1* promoter methylation was also observed in 83.3% of Stage 0 breast cancer patients, 77.1% of patients with smaller-sized tumors and 74.3% of patients without lymph node metastasis (Table 3). Hypermethylation of circulating *CCDC181* and *GCM2* was significantly associated with overexpression of the proliferation marker Ki-67 (Table 3, *p* = 0.012 and 0.021).

### 3.4. Automatic Detection of Circulating Methylated CCDC181, GCM2 and ITPRIPL1 in Plasma Samples from Taiwanese Breast Cancer Patients

To improve the sensitivity and stability of the detection, we increased the plasma sample to 1.6 mL and established an automatic protocol for the extraction of circulating cfDNA, bisulfite conversion and qMSP. We tested the assays in two automatic systems: a column-based system using LabTurbo 24C and the beads-based KingFisher™ Duo Prime Purification System. LabTurbo 24C was used to extract the ccfDNA. The methylation data in each well were similar and provided equivalent results to those obtained with the automatic process (Figure 4A). The sensitivity of the circulating methylated *CCDC181*, *GCM2* and *ITPRIPL1* was 79.0, 62.9 and 76.4%, and the specificity was 64.6, 54.2 and 31.3%, respectively (Table 2).

However, the specificity of the *ITPRIPL1* gene was not satisfactory. Because the MagMAX Cell-Free DNA Isolation Kit can be applied in the automatic KingFisher™ Duo Prime Purification System, we attempted to establish a new automatic protocol to conduct the ccfDNA extraction and subsequently perform bisulfite conversion on the KingFisher™ Duo Prime Purification machine. The repeatability and stability of the process were further confirmed. We assessed 1.6 mL of plasma from six patients in independent wells by parallel testing in two independent KingFisher™ Duo Prime Purification machines. The methylation levels determined using the two Duo Prime machines were similar for parallel testing (Figure 4B).

Using the automatic Duo Prime Purification and bisulfide conversion systems, the sensitivity of circulating methylated *CCDC181*, *GCM2* and *ITPRIPL1* was 65.9, 61.4 and 66.7%, and the specificity was 77.4, 75.7 and 74.1%, respectively (Table 2). The assays using the KingFisher™ Duo Prime purification system and MagMAX Cell-Free DNA Isolation Kit showed better specificity, and thus were majorly used in the following experiments.

### 3.5. Combined Analysis of Methylated Circulating CCDC181, GCM2 and ITPRIPL1

To increase the sensitivity of detection in plasma, methylated circulating *CCDC181*, *GCM2* and *ITPRIPL1* were combined for the calculation. Combined analysis of *CCDC181*, *GCM2* and *ITPRIPL1* was performed by Recursive Partitioning and Regression Trees (RPART) using 57 breast cancer and 134 health plasma samples from the subjects. By splitting of methylation level between the patient and healthy subject samples at a CGI count <0.417 and ≥0.417, the sensitivity was found to be 97.7%, specificity was 85.5%, accuracy was 88.9%, positive predictive value was 72.4% and negative predictive value was 98.9%, and presented as a receiver operating characteristic curve (AUC = 0.955) in the training set; the sensitivity was 92.9%, specificity was 87.5%, accuracy was 89.5%, positive predictive value was 81.3% and negative predictive value was 95.5%, and presented as a receiver operating characteristic curve (AUC = 0.961) in the test set (Table 2 and Figure 4). Breast cancer patients with a positive CGIm value were found mostly among those with lymph node and distance metastasis (96.4% and 100.0%, respectively). A positive CGIm value was also detected in 92.2% of patients with DCIS, 93.4% of patients with invasive ductal carcinoma (IDC), 92.3% of Stage 0 and Stage I patients and 92.2% of patients without distant metastasis (Table 3).

### 3.6. Circulating Methylated CCDC181, GCM2 and ITPRIPL1 in Taiwanese Breast Cancer Patients Were Decreased After Surgery

To further determine the origin of circulating methylated *CCDC181, GCM2* and *ITPRIPL1*, we analyzed whether the DNA methylation between plasma and matched tumor tissues showed a similar pattern. The data indicated that the aberrant circulating hypermethylated *CCDC181*, GCM2 and *ITPRIPL1* detected in the plasma samples were also significantly present in the associated tumor tissues of breast cancer patients (Spearman’s rho = 0.702, 0.497 and 0.634, all *p* < 0.001, Figure 5).

In addition, the aberrant circulating hypermethylated *CCDC181*, *GCM2* and *ITPRIPL1* were dramatically decreased after surgery, especially in earlier disease stages (paired Wilcoxon test; all *p* < 0.001, Figure 6a–c). However, serum levels of CEA and CA 15-3 were not significantly changed after versus before surgery (Figure 6d,e).

### 3.7. The Combination of Clinical Image Data and CGIm Biomarkers for the Early Detection of Breast Cancer

Regular screening mammography (MMG) has been widely used for the early detection of breast cancer in recent years. However, MMG has limitations in the diagnosis of breast diseases for dense breast tissue. This phenomenon is particularly true in Asian women [5,6]. As reported in Table 4, a high percentage of healthy women categorized as BI-RADS 0 (52.6%) were incorrectly referred to the outpatient department of the breast center for further examination. 

Likewise, the healthy women characterized as BI-RADS 2, 3 and 4 (24.4%, 10.3% and 5.1%, respectively) were also erroneously suspected of being breast cancer patients. These results strongly indicate an unreliable diagnosis with MMG alone. As promising biomarkers, circulating methylated *CCDC181*, *GCM2* and *ITPRIPL1* are expected to improve the detection of breast cancer. We performed the analysis combined with BI-RADS (0, 2, 3 and 4) of mammography and circulating methylated *CCDC181*, *GCM2* and *ITPRIPL1* levels. The results were encouraging, with an AUC of 0.918, sensitivity of 0.900 and specificity of 0.772. Even though the false-positive rate was 0.228, the results suggest that there is a significant improvement in breast cancer detection when circulating methylated biomarkers are used.

In comparison to MMG, ultrasonography has been suggested to be a more precise screening approach for the detection of breast lesions in Asian women, especially younger women with dense breasts [4]. However, misclassification still occurs for some breast cancer cases characterized as BI-RADS 3 (4.5%) by ultrasonography who were suspected to be healthy women, as well as healthy women characterized as BI-RADS 4a (25.4%) by ultrasonography who were suspected to be breast cancer patients (Table 4). Particularly, the logistic regression analysis revealed that the false-positive rate was as high as 0.565 with only BI-RADS 3 and 4a. To further investigate whether the use of circulating methylated *CCDC181*, *GCM2* and *ITPRIPL1* biomarkers could improve the detection of breast cancers, we combined the ultrasonography BI-RADS (3 and 4a) and circulating methylated *CCDC181*, *GCM2* and *ITPRIPL1* levels. The results are presented in Table 4. With the inclusion of gene biomarkers, the classification accuracy increased to 0.909 with a corresponding AUC of 0.954, sensitivity of 0.920 and specificity of 0.906. This result indicates that these gene biomarkers contributed significantly to classification and are important markers for the early detection of breast cancers.

## 4. Discussion

Early detection of breast cancer reduces the mortality rates. However, there are no noninvasive and highly sensitive biomarkers for the detection of early breast cancer. Unlike individual mutations that exist only in a subset of tumors, unique DNA methylation patterns are universally present in cells of a common type and therefore may be ideal biomarkers [83]. Promoter hypermethylation of certain genes appears to be an early event in breast cancer carcinogenesis, and this process is mediated through the silencing of tumor suppressor genes involved in cell cycle regulation, DNA repair, transformation, signal transduction, adhesion and metastasis [84]. A previous successful study validated this in advanced later stages of breast cancer (Stage II–IV) [83]. There is little evidence showing that hypomethylated ccfDNA genes could work well as an early prediction biomarker for cancer [85]. It is thus doubtful whether circulating hypomethylated ccfDNA of genes could work well as early prediction biomarkers for breast cancer. Therefore, we focused on the identification of promotor hypermethylation of genes in breast cancer patients from a cohort of Western patients from the TCGA database as well as a Taiwanese cohort in this study. To identify novel potential biomarkers in breast cancer patients, we analyzed a genome-wide methylation array in Taiwanese and TCGA breast cancer patients. Aberrant methylation patterns of the *CCDC181, GCM2, ITPRIPL1, ENPP2, LOC643719, ZNF177* and *ADCY4* genes were identified and further evaluated in paired tumor and normal tissues of breast cancer patients. The methylation patterns of the *CCDC181, GCM2, ITPRIPL1, ENPP2, LOC643719, ZNF177* and *ADCY4* genes were further determined in plasma samples from healthy and early breast cancer patients. Although *RASSF1* has been reported to be hypermethylated in breast cancer patients, it showed a similar pattern in breast cancer patients from both the Taiwan and Western cohorts and was used as the reference control. However, poorer sensitivity or specificity in plasma for the early detection of breast cancer was found for the *LOC643719*, *ENPP2*, *ADCY4* and *RASSF1* genes. In addition, the *LOC643719*, *ZNF177*, *ENPP2*, *ADCY4* and *RASSF1* genes that revealed high methylation patterns in several types of normal tissues from the TCGA cohort were excluded. Finally, the most breast cancer-specific biomarkers, circulating methylated *CCDC181, GCM2* and *ITPRIPL1* (CGIm), were identified in patients with early-stage breast cancer. To improve the repeatability and efficiency of CGIm, an automatic analysis process of methylated ccfDNA was created, and the repeatability was evaluated. The sensitivity and specificity of CGIm for the early detection of breast cancer were 93.9% and 92.0%, respectively. The circulating methylated *CCDC181, GCM2* and *ITPRIPL1* could be significantly decreased after surgery. In addition, we found that the CGIm analysis could be combined with mammography and ultrasound to facilitate the early detection of breast cancer with a sensitivity of 92.0% and specificity of 92.0%.

The functions of the *CCDC181, GCM2, ITPRIPL1, ENPP2, LOC643719, ZNF177* and *ADCY4* genes are described in Appendix A [77,78,79,80]. *GCM2* is a gene encoding a transcription factor that is required for parathyroid development. Mutation of the C-terminal conserved inhibitory domain of *GCM2* can cause primary hyperparathyroidism [80]. Aberrant methylation of *CCDC181* has been reported in patients with lung and prostate cancer [86,87]. *CCDC181* methylation was also found in breast cancer biopsy specimens purified by laser capture microdissection, and it was selected as a fraction marker [82]. In our Taiwanese cohort, hypermethylation of *CCDC181* in tumors was also found in 66.7% of endometrial cancer samples compared with adjacent normal tissues (Appendix A).

Similar results were also observed in the TCGA cohort (Table 3). Although aberrant methylation of *CCDC181* was found in lung and prostate cancer in previous studies and in endometrial cancer in Taiwanese cohort, aberrant methylation of *CCDC181* was a good cancer biomarker with high sensitivity (81.5%) in breast cancer. Combined analysis of high specificity breast cancer biomarker *GCM2* and *ITPRIPL1* can improve the sensitivity and specificity for breast cancer prediction. Therefore, aberrant methylation of *CCDC181*, which is prevalent in female cancer, could serve as a general female tumor marker. *GCM2* is more specific to breast cancer. However, hypermethylation of *ITPRIPL1* has been shown in lung cancer patients. Therefore, if hypermethylation of CGIm is found in patients who are diagnosed without breast cancer, it is suggested that they consult with gynecological or thoracic physicians or oncologists.

The largest fraction of ccfDNA in patients with cancer was derived from the tumor tissue at the site of origin [88]. The bulk of ccfDNA originates from apoptosis, necrosis and active cellular secretion cells in the tumor tissues, tumor microenvironment, cells destroyed under hypoxic conditions or from cells involved in the antitumor response [89]. ccfDNA is often present in the plasma of patients with cancer without detectable CTCs [90]. In this study, we verified the origin of these cell-free circulating hypermethylated DNA fragments by detecting the methylation levels in breast cancer patients before and after surgery. The levels of methylated circulating *CCDC181, GCM2* and *ITPRIPL1* can decline in most patients after surgery, especially in the earlier stages of the disease, such as Stage 0 and Stage I (Figure 6). In addition, monitoring of breast cancer patients after surgical treatment using only the CA-153 tumor marker is insufficient, as shown in a previous study [19]. CA153 and CEA also cannot decrease following surgery. The data suggested that methylated circulating *CCDC181, GCM2* and *ITPRIPL1* were mostly derived from primary tumors and could serve as new postsurgical monitoring biomarkers for the detection of residual tumors, with the exception of primary breast tumors. Although the circulating methylated *ITPRIPL1* gene was not completely diminished after surgery in few patients with earlier-stage tumors, methylated circulating *CCDC181* and *GCM2* were found in most patients with later-stage disease after surgery, suggesting that the origins of residual methylated circulating *CCDC181, GCM2* and *ITPRIPL1* may derive from tumors with microinvasions to vascular or lymph vessels, especially in Stage II and III patients. In addition, it is difficult to monitor the residual tumor cells in the body after surgery in the current follow-up system. Therefore, the detection of methylated circulating *CCDC181, GCM2* and *ITPRIPL1* may facilitate monitoring of the surgical treatment response and assessment of treatment efficiency after chemo, radio, hormone and target therapies.

Although ccfDNA testing is a noninvasive, fast, easily repeatable and sensitive liquid biopsy for cancer detection, it is challenging to repeat due to the extremely low concentration and fragmentation of ccfDNA from whole blood in earlier stages. Therefore, we attempted to link ccfDNA extraction and bisulfite conversion in one machine in an automatic process, with a subsequent transfer to qPCR. First, we performed the analysis using the automatic LabTurbo system. However, the bisulfite conversion efficiency was not satisfactory due to the semi-open heating tube in the automatic machine. Next, we used the KingFisher™ Duo Prime Purification system. The new protocol enabled us to speed up the process without interruption of the steps, and the samples were transferred within 5 h to complete all the steps. The automatic process reduces the chances for human error, helping to ensure consistency, adherence to processes, compliance and increased security by eliminating mistakes. The final results indicated that the use of methylated circulating *CCDC181, GCM2* and *ITPRIPL1* as biomarkers demonstrated good sensitivity (93.9%) and specificity (92.0%). However, the high cost of consumables for the automatic machine may impede the promotion of the automatic protocol. In addition, the fact that relative DNA methylation assay by qMSP was used to analyze the DNA methylation levels rather than absolute DNA methylation assays is the limitation of this study. A relative DNA methylation assay by qMSP is not a good replacement for absolute assays. However, carefully selected, designed and validated qMSP assays can cost-effectively detect trace levels of methylated DNA against an excess of unmethylated DNA [91], especially in hundreds of limited clinical tissue samples or plasma circulating cell-free DNA from patients with early stage breast cancer.

There is no clinically useful, noninvasive biomarker for early detection of breast cancer. Previous studies reported that the circulating cell-free DNA-based epigenetic assay can detect early breast cancer; however, most previous studies used a smaller plasma sample size [16,44,92], did not carry out clinical blood validation [46,93], carried out detection using nipple fluid, which was not easily available in most women [40,94], or used the higher-cost Illumina HumanMethylation450 BeadChip [46,93,95], which may be difficult to use as a universal method in the clinic. Evidence has shown that mammography and ultrasonography support the early diagnosis of breast cancer and reduced breast cancer mortality. Regular screening mammography has been suggested to be associated with improvements in the relative survival of breast cancer patients in recent years. However, the sensitivity of mammography is reduced in women with dense breast tissue [96]. Ultrasonography has been reported to be more precise than mammography for the detection of breast lesions in Asian women, especially for younger women with dense breast tissue [97,98]. However, these techniques have some detection limitations, revealing a high risk for false-negative and false-positive results and requiring experienced radiologists, which significantly affects the sensitivity and specificity. We focused on populations for which there is difficulty with defining disease—BI-RADS 0, 3 and 4 for mammography and 3 and 4a for ultrasonography—and then conducted a combined analysis using methylated circulating *CCDC181, GCM2* and *ITPRIPL1* levels. The data showed a highly improved sensitivity and specificity for the early prediction of breast cancer, especially when combined with ultrasonography. Therefore, we suggested that the methylation pattern of circulating *CCDC181, GCM2* and *ITPRIPL1* could be combined with mammography or/and ultrasonography and applied for the early prediction of breast cancer. The findings may help to provide recommendations for follow-up or further clinical decision making (biopsy or imaging follow-up).

## 5. Conclusions

Aberrant methylation patterns of the *CCDC181, GCM2* and *ITPRIPL1* genes are highly prevalent in Western and Taiwanese breast cancer patients, suggesting that they are potential biomarkers for early prediction in Western and Asian countries. The results can be combined with breast imaging data to provide a higher accuracy, sensitivity and specificity for early breast cancer detection. In addition, they may also be applied to monitor the surgical treatment response and assess the treatment efficiency followed auxiliary therapy.

## Figures and Tables

**Figure 1 cancers-13-01375-f001:**
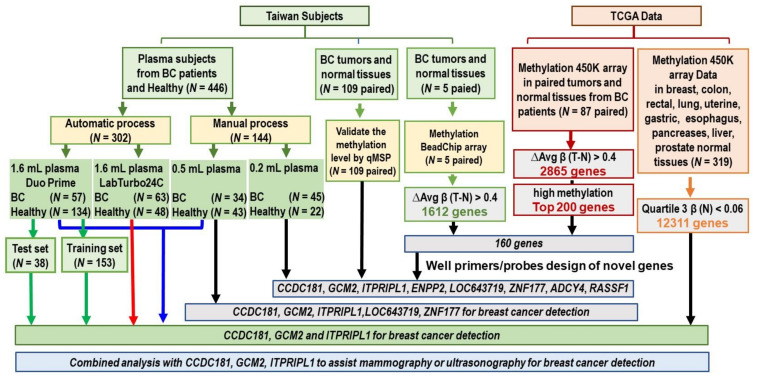
Flowchart of the gene selection, study design, datasets and specimens used (i.e., the criteria and steps used for gene selection). For each step, the sample types and number of samples used for the analyses are indicated. ∆Avg β, β value in the tumor −β value in normal tissue; β, the DNA methylation level ranged from 0 (unmethylated) to 1 (completely methylated); BC, breast cancer; EPIC, Infinium MethylationEPIC array; methylation450K array, Infinium HumanMethylation450 BeadChip array; qMSP, quantitative methylation-specific PCR.

**Figure 2 cancers-13-01375-f002:**
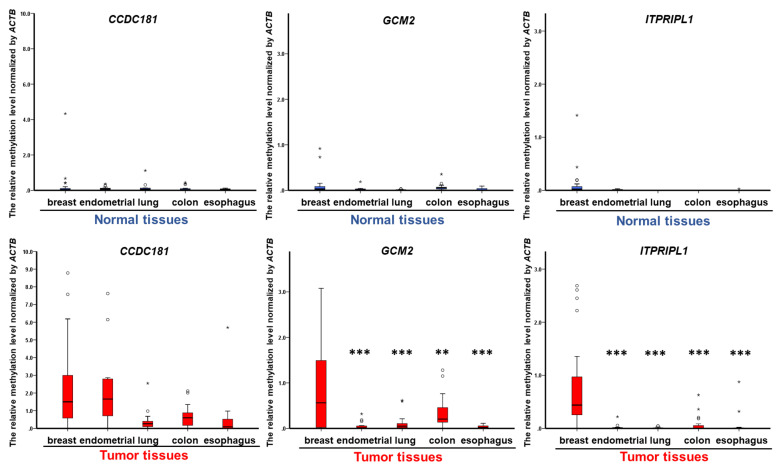
The boxplots for the DNA methylation levels of *CCDC181*, *GCM2* and *ITPRIPL1* in breast cancers and adjacent normal tissues from Taiwanese breast cancer patients. The assays were analyzed by qMSP and included 109 paired breast cancer, 24 paired colon cancer, 16 paired esophageal cancer, 33 paired lung cancer and 15 paired endometrial cancer tissues samples. An independent *t*-test was used to analyze the differences between breast cancer and other cancers samples. ** *p* ≤ 0.01 and *** *p* ≤ 0.001.

**Figure 3 cancers-13-01375-f003:**
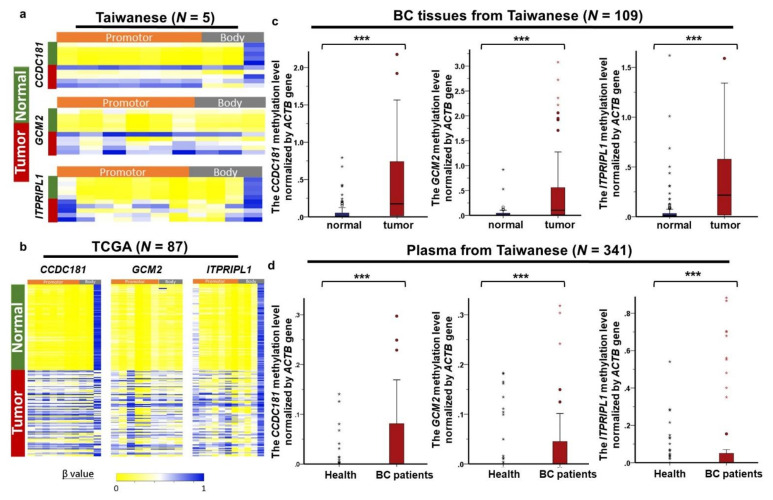
The methylation level of *CCDC181*, *GCM2* and *ITPRIPL1* in tissues and plasma specimens. Differentially methylated CpG heatmap of target genes in five paired Taiwanese BC patients (**a**) and 87 paired samples from the TCGA BC dataset (**b**). Methylation levels (average β values) at differentially methylated loci were identified using Illumina methylation array-based assays. (**c**) The box plot of the target gene methylation levels in 109 BC tumors and paired adjacent normal tissues. A paired Wilcoxon test was used to calculate group differences. (**d**) The box plot of target gene methylation levels in plasma of 141 BC and 200 healthy cases. A Mann–Whitney *U* test was used to calculate group differences. *** all *p* ≤ 0.001.

**Figure 4 cancers-13-01375-f004:**
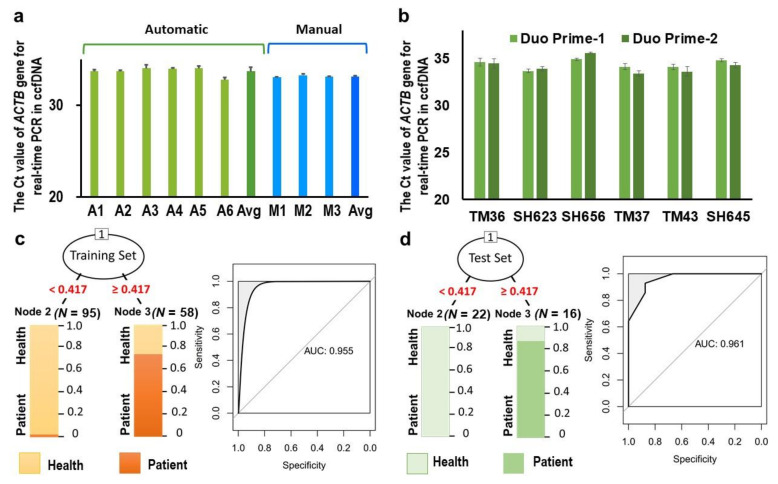
Stability and clinical validation of the automatic detection process for circulating methylated ccfDNA. (**a**) The PCR Ct value of the relative ccfDNA methylation level of the ACTB gene was used to compare the repeatability between the automatic process and manual process. (**b**) The PCR Ct value of the relative ccfDNA methylation level of the ACTB gene was used to compare the stability of the process in independent wells by parallel testing in two independent KingFisher™ Duo Prime Purification machines. (**c**) The decision tree model and ROC curve for the prediction of breast cancer in the training set by Recursive Partitioning and Regression Trees. (**d**) The decision tree model and ROC curve for the prediction of breast cancer in the test set by Recursive Partitioning and Regression Trees.

**Figure 5 cancers-13-01375-f005:**
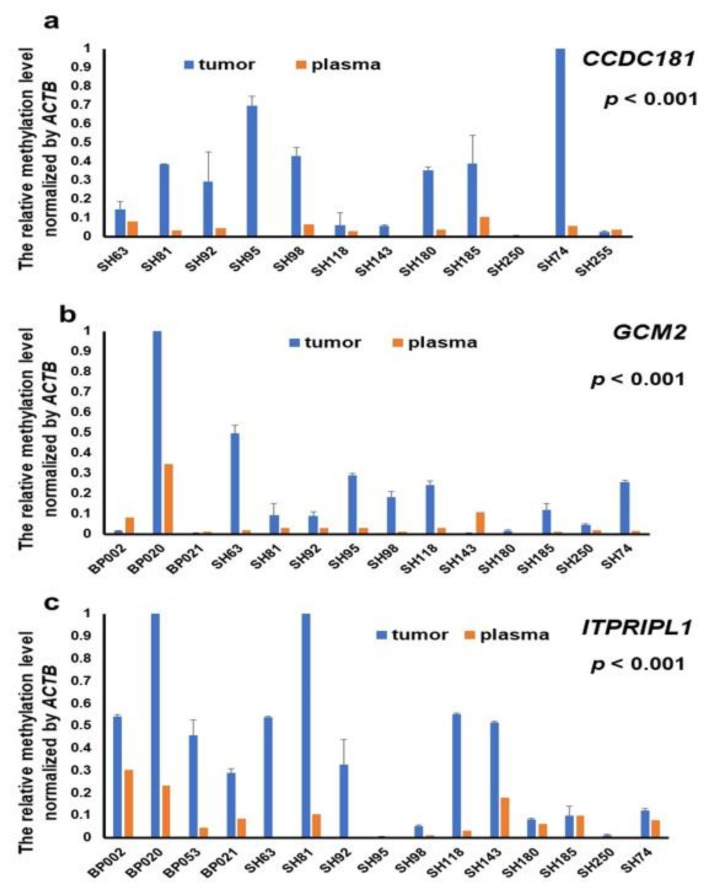
Consistency of the methylation levels of *CCDC181*, *GCM2* and *ITPRIPL1* between tissues and plasma in the same patients are shown. Representative figures of the target gene methylation levels determined by qMSP in BC tumor tissues and matched plasma in the same BC patients. Experiments were performed with three technical replicates. Bar charts for *CCDC181* (**a**), *GCM2* (**b**), and *ITPRIPL1* (**c**) are shown. A Spearman’s rank correlation was used to calculate the correlation between two groups.

**Figure 6 cancers-13-01375-f006:**
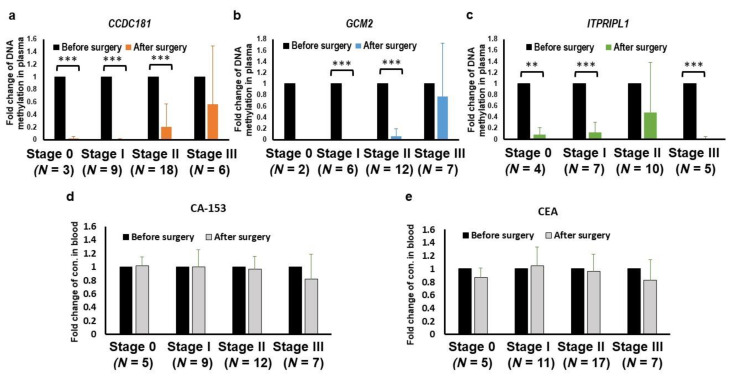
Changes in aberrant circulating hypermethylated *CCDC181*, *GCM2* and *ITPRIPL1* before and after surgery at different stages. Bar charts for *CCDC181* (**a**), *GCM2* (**b**), *ITPRIPL1* (**c**), CA-153 (**d**), and CEA (**e**) are shown; ** *p* ≤ 0.001; *** *p* ≤ 0.001.

**Table 1 cancers-13-01375-t001:** Candidate gene promoter methylation in relation to clinical parameters in tumors of Taiwanese breast cancer ^1^.

Characteristics Methylation ^2^		*CCDC181*		*GCM2*		*ITPRIPL1*			*LOC643719*		*ZNF177*		*ENPP2*		*ADCY4*		*RASSF1*	
		High	Low	High	Low	High	Low		High	Low	High	Low	High	Low	High	Low	High	Low
	*N* ^3^	%	%	%	%	%	%	*N* ^3^	%	%	%	%	%	%	%	%	%	%
Overall	109	81.5	18.5	70.4	29.6	79.6	20.4	55	66.7	33.3	83.0	17.0	77.4	22.6	71.7	28.3	73.6	26.4
Age																		
≤45	16	68.8	31.2	68.8	31.2	81.3	18.7	9	66.7	33.3	100.0	0.0	77.8	22.2	66.7	33.3	77.8	22.2
>45	87	83.9	16.1	70.1	29.9	81.6	18.4	46	54.3	45.7	76.1	23.9	73.9	26.1	69.0	30.4	69.0	30.4
Type																		
DCIS	1	100.0	0.0	100.0	0.0	0.0	100.0	0	0.0	0.0	0.0	0.0	0.0	0.0	0.0	0.0	0.0	0.0
IDC	103	82.4	17.6	70.6	29.4	81.6	18.4	41	65.9	34.1	83.7	16.3	81.6	18.4^0.003^	75.5	24.5^0.012^	73.5	26.5
ILC	2	100.0	0.0	50.0	50.0	50.0	50.0	2	100.0	0.0	100.0	0.0	100.0	0.0	100.0	0.0	100.0	0.0
Others	3	33.3	66.7	66.7	33.3	66.7	33.3	3	66.7	33.3	66.7	33.3	0.0	100.0	0.0	100.0	33.3	66.7
Grade																		
Score 1–2	55	83.6	16.4	80.0	20.0^0.017^	83.6	16.4	29	68.4	31.6	82.6	17.4	73.9	26.1	69.6	30.4	78.3	21.7
Score 3	48	79.2	20.8	58.3	41.7	79.2	20.8	23	69.2	30.8	86.2	13.8	83.3	16.7	75.9	24.1	69.0	31.0
Tumor Stage																		
0, I and II	72	72.9	20.8	69.4	30.6	82.2	17.8	20	66.7	33.3	75.0	25.0	81.8	18.2	71.9	28.1	71.9	28.1
III and IV	29	86.2	13.8	72.4	27.6	78.6	21.4	32	72.2	27.8	95.0	5.0	70.0	30.0	70.0	30.0	70.0	30.0
Tumor Size																		
T0–T2	81	80.2	19.8	69.1	30.9	80.5	19.5	41	64.7	35.3	78.0	22.0	75.6	24.4	70.7	29.3	70.7	29.3
T3–T4	20	85.0	15.0	75.0	25.0	84.2	15.8	12	81.8	18.2	100.0	0.0	83.3	16.7	75.0	25.0	75.0	25.0
Lymph node																		
N = 0	45	80.0	20.0	62.2	37.8	77.8	22.2	30	73.1	26.9	87.1	12.9	77.4	22.6	77.4	22.6	71.0	29.0
N > 0	55	85.5	14.5	72.7	27.3	83.6	16.4	19	66.7	33.3	80.0	20.0	76.2	23.8	65.0	35.0	70.0	30.0
ER																		
Negative	32	75.0	25.0	81.3	18.8	75.0	25.0	14	67.6	32.4	85.0	15.0	77.5	22.5	75.0	25.0	72.5	27.5
Positive	70	87.1	12.9	67.1	32.9	82.9	17.1	40	66.7	33.3	78.6	21.4	78.6	21.4	64.3	35.7	71.4	28.6
PR																		
Negative	45	77.8	22.2	80.0	20.0	82.2	17.8	23	65.4	34.6	83.9	16.1	74.2	25.8	74.3	25.8	74.2	25.8
Positive	57	87.7	12.3	64.9	35.1	78.9	21.1	31	70.0	30.0	82.6	17.4	82.6	17.4	69.6	30.4	69.6	30.4
HER2 ^4^																		
Negative	44	79.5	20.5	61.4	38.6^0.031^	68.9	31.1^0.011^	36	78.6	21.4	88.2	11.8	94.1	5.9	82.4	17.6	82.4	17.6
Positive	57	86.0	14.0	80.7	19.3	89.3	10.7	17	61.3	38.7	80.6	19.4	69.4	30.6	66.7	33.3	69.4	30.6
Ki-67 ^5^																		
High	28	82.1	17.9	64.3	35.7	72.4	27.6	10	75.8	24.2	90.0	10.0^0.041^	87.5	12.5	80.0	20.0	72.5	27.5
Low	67	86.6	13.4	73.1	26.9	85.1	14.9	40	60.0	60.0	60.0	40.0	30.0	70.0	50.0	50.0	60.0	40.0

^1^ These results were analyzed by the Pearson *X^2^* test. Significant *p* values are indicated by superscripts; ^2^ The promoter methylation level in breast tumors that was 2-fold higher than in adjacent normal breast tissues was defined as hypermethylation; ^3^ Because of the limited amount DNA samples from breast tissues, the DNA methylation levels for *LOC643719*, *ZNF177*, *ENPP2*, *ADCY4* and *RASSF1* were only analyzed in 55 breast cancer tissues samples; the DNA methylation level for the *CCDC181*, *GCM2* and *ITPRIPL1* genes was analyzed in 109 patients. For some categories, the number of samples (*N*) was lower than the overall number analyzed because clinical data were unavailable for those samples; ^4^ Overexpression of HER2 was defined as immunohistochemical staining of 2+ or higher; ^5^ Ki-67 values of greater than 14% in breast tumors were defined as overexpression.

**Table 2 cancers-13-01375-t002:** Predictive value of the DNA methylation of the candidate genes for the early prediction of breast cancer.

Manual in 200 μL Plasma ^1^				
*N* = 67	Sensitivity (%)	Specificity (%)	PPV (%)	NPV (%)
*CCDC181*	20.0	100.0	100.0	25.0
*GCM2*	23.3	100.0	100.0	34.3
*ITPRIPL1*	60.0	83.3	85.7	55.6
*ZNF177*	25.6	64.3	66.7	23.7
*LOC643719*	28.6	90.0	88.9	31.0
*RASSF1*	14.3	33.3	33.3	14.3
*ENPP2*	100.0	0.0	62.5	0.0
*ADCY4*	89.3	20.0	75.8	40.0
**Manual in 500 μL plasma ^2^**				
*N* = 77	Sensitivity (%)	Specificity (%)	PPV (%)	NPV (%)
*CCDC181*	75.0	77.8	75.0	77.8
*GCM2*	41.7	100.0	100.0	73.0
*ITPRIPL1*	79.2	96.3	95.0	83.9
*ZNF177*	62.5	85.2	78.9	71.9
*LOC643719*	16.6	96.7	66.7	74.4
**Automatic extraction by LabTurbo 24C in 1.6 mL of plasma ^3^**				
*N* = 111	Sensitivity (%)	Specificity (%)	PPV ^4^ (%)	NPV (%)
*CCDC181*	79.0	64.6	74.2	70.5
*GCM2*	62.9	54.2	63.9	53.1
*ITPRIPL1*	76.4	31.3	56.0	53.6
**Automatic extraction/bisulfite conversion by KingFisher™ Duo Prime in 1.6 mL of plasma ^5^**				
*N* = 191	Sensitivity (%)	Specificity (%)	PPV (%)	NPV (%)
*CCDC181*	65.9	77.4	52.7	85.6
*GCM2*	61.4	75.7	49.1	83.7
*ITPRIPL1*	66.7	74.1	49.1	85.6
	**Training set**			
*N* = 153	Sensitivity (%)	Specificity (%)	PPV (%)	NPV (%)
CGIm ^6^	97.7	85.5	72.4	98.9
	**Test set**			
*N* = 38	Sensitivity (%)	Specificity (%)	PPV (%)	NPV (%)
CGIm	92.9	87.5	81.3	95.5

^1^ The results were tested in 22 plasma samples of healthy and 45 plasma samples of breast cancer patients; ^2^ The results were tested in 43 plasma samples of healthy and 34 plasma samples of breast cancer patients; ^3^ The results were validated in 48 plasma samples of healthy and 63 plasma samples of breast cancer patients; ^4^ PPV: positive predictive value (precision); NPV: negative predictive value; ^5^ The results were trained on 110 plasma samples of healthy and 43 plasma samples of breast cancer patients, tested in 24 plasma samples of healthy and 14 plasma samples of breast cancer patients; ^6^ CGIm, derived from a combination analysis of methylated circulating *CCDC181*, *GCM2* and *ITPRIPL1*, defined when the CGIcount was less than 0.417. The CGIcount was calculated by Recursive Partitioning and Regression Trees (RPART).

**Table 3 cancers-13-01375-t003:** *CCDC181, GCM2* and *ITPRIPL1* promoter methylation in relation to clinical parameters in plasma of Taiwanese breast cancer patients ^1^.

Characteristics Methylation	*N*	*CCDC181*		*GCM2*		*ITPRIPL1*		CGIm *^3^*	
		Negative	Positive	Negative	Positive	Negative	Positive	Negative	Positive
		*N* (%)	*N* (%)	*N* (%)	*N* (%)	*N* (%)	*N* (%)	*N* (%)	*N* (%)
**Overall ^2^**		49 (31.6)	106 (68.4)	79 (45.7)	94 (54.3)	35 (25.0)	105 (75.0)	10 (7.1)	131 (92.9)
**Age**									
<30	8	2 (25.0)	6 (75.0)	2 (25.0)	6 (75.0)	2 (25.0)	6 (75.0)	0 (0.0)	8 (100.0)
40–49	38	12 (31.6)	26 (68.4)	13 (34.2)	25 (65.8)	13 (34.2)	25 (65.8)	3 (7.7)	36 (92.3)
50–59	42	11 (26.8)	30 (73.2)	15 (36.6)	26 (63.4)	7 (20.0)	28 (80.0)	3 (7.3)	38 (92.7)
60–69	50	10 (20.4)	39 (79.6)	21 (42.9)	28 (57.1)	12 (25.5)	35 (74.5)	3 (6.1)	46 (93.9)
≧70	3	0 (0.0)	3 (100.0)	0 (0.0)	3 (100.0)	0 (0.0)	2 (100.0)	0 (0.0)	3 (100.0)
**Type**									
DCIS	15	5 (35.7)	9 (65.3)^0.028^	6 (42.9)	8 (57.1)	2 (15.4)	11 (84.6)	1 (7.1)	13 (92.9)
IDC	177	36 (30.0)	84 (70.0)	64 (46.7)	73 (53.3)	28 (26.4)	78 (73.6)	7 (6.6)	99 (93.4)
ILC	5	0 (0.0)	5 (100.0)	1 (20.0)	4 (80.0)	2 (40.0)	3 (60.0)	0 (0.0)	5 (100.0)
Others	3	3 (100.0)	0 (0.0)	3 (100.0)	0 (0.0)	0 (0.0)	3 (100.0)	1 (33.3)	2 (66.7)
**Grade**									
Grade 1	19	2 (28.6)	5 (71.4)	9 (69.2)	4 (30.8)	1 (25.0)	3 (75.0)	0 (0.0)	4 (100.0)
Grade 2	43	10 (38.5)	16 (61.5)	19 (69.3)	12 (38.7)	7 (31.8)	15 (68.2)	4 (17.4)	19 (82.6)
Grade 3	47	11 (51.4)	10 (47.6)	16 (57.1)	12 (42.9)	2 (10.5)	17 (89.5)	1 (8.3)	11 (91.7)
**Tumor Stage**									
Stage 0	17	6 (46.2)	7 (53.8)	8 (53.3)	7 (46.7)	2 (16.7)	10 (83.3)	1 (7.7)	12 (92.3)
Stage I	45	12 (40.0)	18 (60.0)	17 (51.5)	16 (48.5)	7 (29.2)	17 (70.8)	2 (7.7)	24 (92.3)
Stage II	103	16 (21.6)	58 (78.4)	37 (44.6)	46 (55.4)	16 (23.9)	51 (76.1)	6 (8.7)	63 (91.3)
Stage III	34	9 (37.5)	15 (62.5)	13 (46.4)	15 (53.6)	6 (26.1)	17 (73.9)	0 (0.0)	19 (100.0)
Stage IV	7	0 (0.0)	6 (100.0)	2 (33.3)	4 (66.7)	1 (16.7)	5 (83.3)	0 (0.0)	6 (100.0)
**Tumor Size**									
T0–T2	161	36 (32.7)	74 (67.3)	61 (48.4)	65 (51.6)	22 (22.9)	74 (77.1)	8 (8.1)	91 (91.9)
T3–T4	35	7 (25.0)	21 (75.0)	13 (43.3)	17 (56.7)	10 (37.0)	17 (63.0)	2 (8.0)	23 (92.0)
**Lymph node**									
N = 0	105	24 (32.0)	51 (68.0)	40 (48.2)	43 (51.8)	18 (25.7)	52 (74.3)	8 (11.4)	62 (88.6)
N > 0	91	19 (29.2)	46 (70.8)	34 (45.3)	41 (54.7)	14 (25.5)	41 (74.5)	2 (3.6)	53 (96.4)
**Metastasis**									
No	188	43 (32.8)	88 (67.2)	71 (48.0)	77 (52.0)	29 (25.2)	86 (74.8)	9 (7.8)	107 (92.2)
Yes	7	0	6 (100.0)	2 (33.3)	4 (66.7)	1 (16.7)	5 (83.3)	0	6 (100.0)
**ER**									
Negative	70	16 (37.2)	27 (62.8)	26 (48.1)	28 (51.9)	8 (21.1)	30 (78.9)	2 (5.4)	35 (94.6)
Positive	140	31 (28.4)	78 (71.6)	52 (44.8)	64 (55.2)	27 (27.3)	72 (72.7)	8 (8.0)	92 (92.0)
**PR**									
Negative	77	16 (31.4)	35 (68.6)	24 (41.4)	34 (58.6)	11 (23.4)	36. (76.6)	3 (7.0)	40 (93.0)
Positive	133	31 (30.7)	70 (69.3)	54 (48.2)	58 (51.8)	24 (26.7)	66 (73.3)	7 (7.4)	87 (92.6)
**HER2 ^4^**									
Negative	142	31 (31.3)	68 (68.7)	52 (45.6)	62 (54.4)	24 (27.9)	62 (72.1)	6 (6.7)	83 (93.3)
Positive	44	9 (31.0)	20 (69.0)	16 (50.0)	16 (50.0)	3 (10.7)	25 (89.3)	0 (0.0)	25 (100.0)
**TNBC**									
No	136	30 (26.1)	85 (73.9)	52 (42.3)	71 (57.7)	29 (27.6)	76 (72.4)	8 (7.1)	104 (92.9)
Yes	70	14 (42.4)	19 (57.6)	24 (55.8)	19 (44.2)	6 (21.4)	22 (78.6)	2 (9.5)	19 (90.5)
**Ki-67 ^5^**									
Low	64	17 (47.2)	19 (52.8)^0.012^	29 (59.2)	20 (40.8)^0.021^	11 (36.7)	19 (63.3)	2 (6.5)	29 (93.5)
High	121	22 (24.4)	68 (75.6)	37 (38.9)	58 (61.1)	20 (24.7)	61 (75.3)	7 (8.6)	74 (91.4)

^1^ These results were analyzed by the Pearson *X^2^* test. Significant *p* values are denoted by superscripts; ^2^ For some categories, the number of samples (*N*) was lower than the overall number analyzed since (1) due to plasma volume limitation, some genes were analyzed in different patients; and (2) clinical data were unavailable for some samples; ^3^ The methylation of *CCDC181*, *GCM2* and *ITPRIPL1* was analyzed in the same patients at the same time. CGIm positive was defined by the results of the combined analysis of *CCDC181*, *GCM2* and *ITPRIPL1* methylation using Decision Trees in R with the RPART model; ^4^ Overexpression of HER2 was defined as immunohistochemistry staining was of 2+ or higher; ^5^ Ki-67 values of greater than 14% in breast tumors were defined as overexpression.

**Table 4 cancers-13-01375-t004:** Combined analysis of circulating methylated biomarkers, mammography and ultrasound.

Mammography								
BI-RADS2	*N*	BI-RADS 0	BI-RADS 1	BI-RADS 2	BI-RADS 3	BI-RADS 4	BI-RADS 5	
Health	78	41 (52.6)	6 (7.7)	19 (24.4)	8 (10.3)	4 (5.1)	0	
BC patients	45	23 (51.1)	0 (0.0)	8 (17.8)	7 (15.6)	5 (11.1)	2 (4.4)	
Including gene biomarkers additionally BI-RADS (0, 2, 3 and 4), *CCDC181*, *GCM2* and *ITPRIPL1*								
Kang’s stepwise method		Accuracy		0.805	AUC		0.918	
		Sensitivity		0.900	Specificity		0.772	
		False-Positive Rate		0.228	False-Negative Rate		0.100	
Breast Ultrasound1								
Healthy BC patients	*N*	BI-RADS 2	BI-RADS 3	BI-RADS 4a	BI-RADS 4b	BI-RADS 4c	BI-RADS 5	BI-RADS 6
	195	94 (48.2)	51 (26.2)	50 (25.6)	0 (0.0)	0 (0.0)	0 (0.0)	0 (0.0)
	111	0 (0.0)	5 (4.5)	44 (39.6)	29 (26.1)	23 (20.7)	8 (7.2)	2 (1.8)
Ultrasound only: BI-RADS 3 and 4a								
Logistic regression		Accuracy		0.545				
		Sensitivity		0.920	Specificity		0.435	
		False-Positive Rate		0.565	False-Negative Rate		0.080	
Including gene biomarkers additionally BI-RADS (3 and 4a), *CCDC181*, *GCM2* and *ITPRIPL1*								
Kang’s stepwise method		Accuracy		0.909	AUC		0.954	
		Sensitivity		0.920	Specificity		0.906	
		False-Positive Rate		0.094	False-Negative Rate		0.080	

The patients were recruited from the outpatient department of the breast center. BI-RADS, Breast Imaging Reporting and Data System.

## Data Availability

The data generated in this study are available from the corresponding author upon reasonable request.

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
