# Peer review of "Automatic Detection of the Circulating Cell-Free Methylated DNA Pattern of GCM2, ITPRIPL1 and CCDC181 for Detection of Early Breast Cancer and Surgical Treatment Response"

_cancers, 2021, doi:10.3390/cancers13061375_

Round 1

Reviewer 1 Report

Manuscript ID: cancers-1107117

Title: Automatic Detection of Circulating Cell-Free Methylated DNA Pattern of GCM2, ITPRIPL1, and CCDC181 for Detection of Early Breast Cancer and Surgical Treatment Response.

This manuscript establishes a relationship between specific DNA methylated genes and breast cancer to define biomarkers for diagnosis, evolution, and early detection of the disease. The study is sound, interesting, and valuable; however, the manuscript needs a thorough revision of the English language because some sentences are difficult to understand or are incomplete. Besides, grammar and punctuation demand revision. Some examples appear at the end of this report. Concerns and corrections follow.

Simple summary and abstract sections should be revised to clarify the reported results and their interpretation and application. For example, the section Background within the abstract commences with the sentence "The aim of this study…" That should appear at the end of the sub-section that may commence with a brief statement concerning the motive or reason to perform this study to continue with "The aim..."

The authors write: "After a literature review, aberrant methylated genes that were reported previously were excluded" (line 300). What do the authors mean by aberrant methylated genes as a criterium for the exclusion? The selection of the candidate genes, line 302, needs a more explicit explanation to include or exclude a particular gene. Is it because only the selected genes were consistently found hypermethylated in both cohorts studied?

Table 2 presents specific methylation patterns of the selected genes in different histological BC types, tumor size and stage, presence of adenopathies, grade, etc. Would it be possible, with the results obtained, to use the methylation patterns in one or several genes as a biomarker of the complex picture (i.e., not only related to a trait such as size, histological type, etc., but to a clinical situation that encompasses many variables; age, BC type, size of the tumor…). Would it be possible to associate the methylation patterns (of one or several genes) to a specific variable relevant to prognosis and treatment?

The reliability and significance of cell-free circulating hypermethylated DNA fragments as an adequate measure need mention and discussion (For example, their origin and stability).

Minor corrections

Abbreviations for the names of the genes studied, and other abbreviations such as TCGA (it appears on line 112) need explanation the first time they appear in the text (as the authors provide in many other abbreviations appearing in this manuscript). They may result familiar for experts but not for all potential readers.

Some words appear cut with hyphens. It may be due to the adjustment of the text to the template form. Please, check. A few examples follow: cancer-spe-cific (line 39), mammogra-phy (line 63), mammogra-phy (line 90), can-cer tissues (line114),

The sentence on lines 139-141 needs revision. Writing a period (.) before "within 2-hour" may help. Please check.

After a literature review, aberrant methylated genes that were reported (line 300). Aberrantly instead of aberrant?

Author Response

Response to Reviewer 1’s comments:

Point 1: Simple summary and abstract sections should be revised to clarify the reported results and their interpretation and application. For example, the section Background within the abstract commences with the sentence "The aim of this study…" That should appear at the end of the sub-section that may commence with a brief statement concerning the motive or reason to perform this study to continue with "The aim..."

Response 1: We thank the reviewer for their suggestions and comments. We have modified the simple summary and abstract sections (Page 1, simple summary line 27-32 and abstract line 33-38).

Point 2: The authors write: "After a literature review, aberrant methylated genes that were reported previously were excluded" (line 300). What do the authors mean by aberrant methylated genes as a criterium for the exclusion? The selection of the candidate genes, line 302, needs a more explicit explanation to include or exclude a particular gene. Is it because only the selected genes were consistently found hypermethylated in both cohorts studied?

Response 2: We thankful the reviewer for the suggestions and comments. A total of 1190 genes were found to be overlapped between 1612 genes in the Taiwanese cohort and 2865 genes in the TCGA cohort. This study focused on the top 200 genes with the highest methylation levels in breast tumors from TCGA. In total, 160 highly methylated genes were found when the genes identified from the Taiwanese and TCGA cohorts were combined (Table S2). A literature review was conducted and the following previously-reported aberrantly methylated genes were excluded: DPF1 [33], DNM3 [16], TLX1 [34], CHST11 [35], CHST3 [36], ALX1 [37], HNF1B [34], EBF1 [38], PITX2 [39], ITGA5 [40], COL11A2 [41], CFTR [42], NR5A2 [43], PRKCB [43], SOSTDC1 [44], DBX1 [45], LHX8 [46], HCK [47], NXPH1 [48], NID2 [49], TAC1 [50], TFAP2B [51], F2RL3 [52], POU4F2 [53], H2AFY [54], GALR1 [55], FOXD3 [56], NT5E [57], CLIP4 [58], ZNF454 [59], TIMP2 [60], GNG4 [61], MIR129-2 [61], TXNRD1 [62], CPXM1 [61], PRDM14 [61], HOXA4 [63], SLITRK1 [64], AHRR [65], NPTX2 [66], KCNK9 [67], C9orf122 [68], CRHR2 [69], BOLL [70], CCDC8 [71], MMP9 [72], C12orf68 [68]. The remaining candidate genes were then studied through sequencing and gene-specific primers/probe design to determine their methylation-specific PCR reactions. The results for seven candidate genes—CCDC181, GCM2, ITPRIPL1, ENPP2, LOC643719, ZNF177, and ADCY4—were successfully validated by sequencing and stable detection in ccfDNA from the plasma of Taiwanese patients with breast cancer. The seven candidate genes, CCDC181, GCM2, ITPRIPL1, ENPP2, LOC643719, ZNF177, ADCY4, were found to be hypermethylated in both the Taiwanese and Western cohorts (Figure 1). RASSF1 has been reported to be hypermethylated in breast cancer and showed a similar pattern to the reference control in breast cancer patients from both the Taiwan and Western cohorts [73]. Therefore, CCDC181, GCM2, ITPRIPL1, ENPP2, LOC643719, ZNF177, ADCY4, and RASSF1 were analyzed in the following assays.

We have added the descriptions in the line 334-358 of result section.

Point 3: Table 2 presents specific methylation patterns of the selected genes in different histological BC types, tumor size and stage, presence of adenopathies, grade, etc. Would it be possible, with the results obtained, to use the methylation patterns in one or several genes as a biomarker of the complex picture (i.e., not only related to a trait such as size, histological type, etc., but to a clinical situation that encompasses many variables; age, BC type, size of the tumor…). Would it be possible to associate the methylation patterns (of one or several genes) to a specific variable relevant to prognosis and treatment?

Response 3: We thank the reviewer for their suggestions and comments. We tried to find a prediction model for specific breast cancer subtypes by analyzing the methylation patterns of multiple genes, such as those found in triple negative breast cancer (TNBC) patients. One of our unpublished regression analyses has been shown to have good predictive value when using the methylation levels of SLC7A4, ADCY4, ENPP2, RASSF1, and LOC643719. The prediction model was calculated using the following formula: TNBC=0.560+(-0.494*SLC7A4) + (0.526*ADCY4) + (0.350*ENPP2) + (0.259*RASSF1) + (-0.289* LOC643719). If >1=TNBC; if <1=other subtypes. The model has not been validated for use with blood samples; therefore, the results are not shown in this manuscript.

Point 4: The reliability and significance of cell-free circulating hypermethylated DNA fragments as an adequate measure need mention and discussion (For example, their origin).

Response 4: We thank the reviewer for their suggestions and comments. The largest fraction of ccfDNA in patients with cancer was derived from the tumor tissue at the site of origin [85]. The bulk of ccfDNA originates from apoptosis, necrosis, and active cellular secretion cells in the tumor tissues, tumor microenvironment, cells destroyed under hypoxic conditions, or from cells involved in the antitumor response [86]. ccfDNA is often present in the plasma of patients with cancer without detectable CTCs [87]. In this study, we verified the origin of these cell-free circulating hypermethylated DNA fragments by detecting the methylation levels in breast cancer patients before and after surgery.

The description has been added to lines 659–665 of the discussion section.

Point 5: Abbreviations for the names of the genes studied, and other abbreviations such as TCGA (it appears on line 112) need explanation the first time they appear in the text (as the authors provide in many other abbreviations appearing in this manuscript). They may result familiar for experts but not for all potential readers.

Response 5: We thank the reviewer for their suggestions and comments. The explanations of the abbreviations have been corrected.

Point 6: Some words appear cut with hyphens. It may be due to the adjustment of the text to the template form. Please, check. A few examples follow: cancer-spe-cific (line 39), mammogra-phy (line 63), mammogra-phy (line 90), can-cer tissues (line114),

Response 6: The error has been corrected.

Point 7: The sentence on lines 139-141 needs revision. Writing a period (.) before "within 2-hour" may help. Please check.

Response 7: We thank the reviewer for their suggestions and comments. We have corrected it.

Point 8: After a literature review, aberrant methylated genes that were reported (line 300). Aberrantly instead of aberrant?

Response 8: We thankful the reviewer for the suggestions and comments. We have corrected it.

References:

  1. Lo, Y.M. Noninvasive prenatal diagnosis: from dream to reality. Clin Chem 2015, 61, 32-37, doi:10.1373/clinchem.2014.223024.
  2. Lo, Y.M.; Chan, K.C.; Sun, H.; Chen, E.Z.; Jiang, P.; Lun, F.M.; Zheng, Y.W.; Leung, T.Y.; Lau, T.K.; Cantor, C.R., et al. Maternal plasma DNA sequencing reveals the genome-wide genetic and mutational profile of the fetus. Sci Transl Med 2010, 2, 61ra91, doi:10.1126/scitranslmed.3001720.
  3. Lou, X.; Hou, Y.; Liang, D.; Peng, L.; Chen, H.; Ma, S.; Zhang, L. A novel Alu-based real-time PCR method for the quantitative detection of plasma circulating cell-free DNA: sensitivity and specificity for the diagnosis of myocardial infarction. Int J Mol Med 2015, 35, 72-80, doi:10.3892/ijmm.2014.1991.
  4. Heitzer, E.; Auer, M.; Hoffmann, E.M.; Pichler, M.; Gasch, C.; Ulz, P.; Lax, S.; Waldispuehl-Geigl, J.; Mauermann, O.; Mohan, S., et al. Establishment of tumor-specific copy number alterations from plasma DNA of patients with cancer. Int J Cancer 2013, 133, 346-356, doi:10.1002/ijc.28030.
  5. Swarup, V.; Rajeswari, M.R. Circulating (cell-free) nucleic acids--a promising, non-invasive tool for early detection of several human diseases. FEBS Lett 2007, 581, 795-799, doi:10.1016/j.febslet.2007.01.051.
  6. Zonta, E.; Nizard, P.; Taly, V. Assessment of DNA Integrity, Applications for Cancer Research. Adv Clin Chem 2015, 70, 197-246, doi:10.1016/bs.acc.2015.03.002.
  7. Laird, P.W. The power and the promise of DNA methylation markers. Nat Rev Cancer 2003, 3, 253-266, doi:10.1038/nrc1045.
  8. Imperiale, T.F.; Ransohoff, D.F.; Itzkowitz, S.H. Multitarget stool DNA testing for colorectal-cancer screening. N Engl J Med 2014, 371, 187-188, doi:10.1056/NEJMc1405215.
  9. Warren, J.D.; Xiong, W.; Bunker, A.M.; Vaughn, C.P.; Furtado, L.V.; Roberts, W.L.; Fang, J.C.; Samowitz, W.S.; Heichman, K.A. Septin 9 methylated DNA is a sensitive and specific blood test for colorectal cancer. BMC Med 2011, 9, 133, doi:10.1186/1741-7015-9-133.

Reviewer 2 Report

In this study authors present a way of detecting cell-free methylated DNA in candidate genes for early detection of breast cancer as a non-invasive biomarker. However, there are some points need to be addressed/reviewed. Please see below for my comments.

In line 28-29, Authors say “There was no useful non-invasive biomarker for early detection of breast cancer”. In discussion part, not only there is no discussion about published non-invasive biomarkers for breast cancer but also the reason that makes this research better compared to published biomarkers.  

Figure 1 is very confusing to follow. In the results section, it is poorly described how these patients were divided into groups based on what and how the criteria for these groups were determined was not described clearly and very confusing to follow.

Line organization is poorly designed in Table S1. If any other researcher wants to order the same primer sets it is difficult follow where it ends and starts.  

In line 300, “After a literature review, aberrant methylated genes that were reported previously were excluded”. Reference is missing and what these genes are not listed anywhere.

In line 301-303, “Then, gene-specific primers/probe testing, sequencing and validation in Taiwanese patients with breast cancer and healthy subjects revealed 7 candidate genes, CCDC181, GCM2, ITPRIPL1, ENPP2, LOC643719, ZNF177, ADCY4, which were hypermethylated in both the Taiwanese and Western cohort (Figure 1). This part is the most confusing part of this manuscript. It is very difficult to follow how these genes were discovered.

In line 304-305, RASSF1 has been reported to be hypermethylated in breast cancer and showed a similar pattern in breast cancer patients in both the Taiwan and Western cohorts as the reference control.” Explanation of why this gene was not added to the other 7 candidate genes was poorly described.  

In line 334, “For some categories, the number of samples (N) was lower than the overall number analyzed because clinical data were unavailable for those samples.” If clinical data is not available, then why these samples are included in the study was not explained.

In line 384, “Few studies have reported the clinical significance and application of aberrant CCDC181, GCM2 and ITPRIPL1 methylation in breast cancer”. Reference is missing.

In lines 512-514, “Even though the false positive rate was somewhat high, the results suggested a significant improvement in breast cancer detection using circulating methylated biomarkers.” First of all, using a word like “somewhat high” doesn’t sound scientific. Secondly, if something has a high false positive rate how trustable is the conclusion as a significant improvement.

In line 562, “Aberrant methylation of CCDC181 has been reported in lung and prostate.” Is it cancer or healthy tissues? If CCDC181 is found in these cancer types, then it means that it is not breast cancer specific biomarker as authors suggested.

Author Response

Response to Reviewer 2 comments:

Point 1: In line 28-29, Authors say “There was no useful non-invasive biomarker for early detection of breast cancer”. In discussion part, not only there is no discussion about published non-invasive biomarkers for breast cancer but also the reason that makes this research better compared to published biomarkers.

Response 1: We apologize for the lack of clarity. We have modified the descriptions as follow: There is no clinically useful non-invasive biomarker for early detection of breast cancer. Previous studies reported that the circulating cell-free DNA-based epigenetic assay can detect early breast cancer; however, most previous studies used a smaller plasma sample size [1-3], did not carry out clinical blood validation [4,5], carried out detection using nipple fluid, which was not easily available in most women [6,7], or used the higher-cost Illumina HumanMethylation450 BeadChip [4,5,8], which may be difficult to use as a universal method in the clinic. The description has been added to lines 700-706 of the discussion section.

Point 2: Figure 1 is very confusing to follow. In the results section, it is poorly described how these patients were divided into groups based on what and how the criteria for these groups were determined was not described clearly and very confusing to follow.

Response 2: We apologize for the lack of clarity. The first patient pool (0.2 mL plasma sample) was obtained from the TMU biobank, used for first biomarkers screening. The patient pool (0.5 mL plasma sample) was recruited from Taipei Medical University (TMU) Hospital, Shuang Ho Hospital, that was used for second biomarkers screening. To enhance the sensitivity and stability of analysis, we performed the assay in the third patient pool (1.6 mL plasma sample). The third patient pool was also recruited from Taipei Medical University (TMU) Hospital and Shuang Ho Hospital.

The description has been added to lines 147–153 of Section 2.1.

Point 3: Line organization is poorly designed in Table S1. If any other researcher wants to order the same primer sets it is difficult follow where it ends and starts.

Response 3: We thank the reviewer for their suggestions and comments. We have corrected the error in Table S1.

Point 4: In line 300, “After a literature review, aberrant methylated genes that were reported previously were excluded”. Reference is missing and what these genes are not listed anywhere.

Response 4: We thankful the reviewer for the suggestions and comments. Aberrantly methylated genes that had been reported previously were excluded: DPF1 [9], DNM3 [1], TLX1 [10], CHST11 [11], CHST3 [12], ALX1 [6], HNF1B [10], EBF1 [13], PITX2 [14], ITGA5 [15], COL11A2 [2], CFTR [16], NR5A2 [4], PRKCB [4], SOSTDC1 [17], DBX1 [18], LHX8 [19], HCK [20], NXPH1 [21], NID2 [22], TAC1 [23], TFAP2B [24], F2RL3 [25], POU4F2 [26], H2AFY [27], GALR1 [28], FOXD3 [29], NT5E [30], CLIP4 [31], ZNF454 [32], TIMP2 [33], GNG4 [34], MIR129-2 [34], TXNRD1 [35], CPXM1 [34], PRDM14 [34], HOXA4 [36], SLITRK1 [37], AHRR [38], NPTX2 [39], KCNK9 [40], C9orf122 [41], CRHR2 [42], BOLL [43], CCDC8 [44], MMP9 [45], and C12orf68 [41].

We have added the references to lines 338–346 of the manuscript.

Point 5: In line 301-303, “Then, gene-specific primers/probe testing, sequencing and validation in Taiwanese patients with breast cancer and healthy subjects revealed 7 candidate genes, CCDC181, GCM2, ITPRIPL1, ENPP2, LOC643719, ZNF177, ADCY4, which were hypermethylated in both the Taiwanese and Western cohort (Figure 1). This part is the most confusing part of this manuscript. It is very difficult to follow how these genes were discovered.

Response 5: We apologize that we didn’t described clear.

A total of 1190 genes were found to be overlapped between 1612 genes in the Taiwanese cohort and 2865 genes in the TCGA cohort. This study focused on the top 200 genes with the highest methylation levels in breast tumors from TCGA. In total, 160 highly methylated genes were found when the genes identified from the Taiwanese and TCGA cohorts were combined (Table S2). A literature review was conducted and the following previously-reported aberrantly methylated genes were excluded: DPF1 [9], DNM3 [1], TLX1 [10], CHST11 [11], CHST3 [12], ALX1 [6], HNF1B [10], EBF1 [13], PITX2 [14], ITGA5 [15], COL11A2 [2], CFTR [16], NR5A2 [4], PRKCB [4], SOSTDC1 [17], DBX1 [18], LHX8 [19], HCK [20], NXPH1 [21], NID2 [22], TAC1 [23], TFAP2B [24], F2RL3 [25], POU4F2 [26], H2AFY [27], GALR1 [28], FOXD3 [29], NT5E [30], CLIP4 [31], ZNF454 [32], TIMP2 [33], GNG4 [34], MIR129-2 [34], TXNRD1 [35], CPXM1 [34], PRDM14 [34], HOXA4 [36], SLITRK1 [37], AHRR [38], NPTX2 [39], KCNK9 [40], C9orf122 [41], CRHR2 [42], BOLL [43], CCDC8 [44], MMP9 [45], and C12orf68 [41]. The remaining candidate genes were then studied through sequencing and gene-specific primers/probe design to determine their methylation-specific PCR reactions. The results for seven candidate genes—CCDC181, GCM2, ITPRIPL1, ENPP2, LOC643719, ZNF177, and ADCY4—were successfully validated by sequencing and stable detection in ccfDNA from the plasma of Taiwanese patients with breast cancer. The seven candidate genes, CCDC181, GCM2, ITPRIPL1, ENPP2, LOC643719, ZNF177, ADCY4, were found to be hypermethylated in both the Taiwanese and Western cohorts (Figure 1). RASSF1 has been reported to be hypermethylated in breast cancer and showed a similar pattern to the reference control in breast cancer patients from both the Taiwan and Western cohorts [46]. Therefore, CCDC181, GCM2, ITPRIPL1, ENPP2, LOC643719, ZNF177, ADCY4, and RASSF1 were analyzed in the following assays.

We have added the descriptions to lines 334–358 of the results section.

Point 6: In line 304-305, RASSF1 has been reported to be hypermethylated in breast cancer and showed a similar pattern in breast cancer patients in both the Taiwan and Western cohorts as the reference control.” Explanation of why this gene was not added to the other 7 candidate genes was poorly described.

Response 6: We thank the reviewer for their suggestions and comments. The explanation has been added to lines 623-628 of the discussion section as follows: The methylation patterns of the CCDC181, GCM2, ITPRIPL1, ENPP2, LOC643719, ZNF177, and ADCY4 genes were further determined in plasma samples from healthy and early breast cancer patients. Although RASSF1 has been reported to be hypermethylated in breast cancer patients, it showed a similar pattern in breast cancer patients from both the Taiwan and Western cohorts and was used as the reference control. However, poorer sensitivity or specificity in plasma for the early detection of breast cancer was found for the LOC643719, ENPP2, ADCY4, and RASSF1 genes. In addition, the LOC643719, ZNF177, ENPP2, ADCY4 and RASSF1 genes were excluded that revealed high methylation patterns in several types of normal tissues from the TCGA cohort.

Point 7: In line 334, “For some categories, the number of samples (N) was lower than the overall number analyzed because clinical data were unavailable for those samples.” If clinical data is not available, then why these samples are included in the study was not explained.

Response 7: We apologize that we didn’t described clear. Because of the limited amount DNA samples from breast tissues, the DNA methylation levels for LOC643719, ZNF177, ENPP2, ADCY4, and RASSF1 were only analyzed in 55 breast cancer tissues samples; the DNA methylation level for the CCDC181, GCM2, and ITPRIPL1 genes was analyzed in 109 patients.

The description has been added to the footnote of Table 1.

Point 8: In line 384, “Few studies have reported the clinical significance and application of aberrant CCDC181, GCM2 and ITPRIPL1 methylation in breast cancer”. Reference is missing.

Response 8: We thank the reviewer for their suggestions and comments. DNA methylation of CCDC181 was reported as a breast cancer cell fraction marker in 2018 [47]. The reference has been cited in line 439-440 of the manuscript.

Point 9: In lines 512-514, “Even though the false positive rate was somewhat high, the results suggested a significant improvement in breast cancer detection using circulating methylated biomarkers.” First of all, using a word like “somewhat high” doesn’t sound scientific. Secondly, if something has a high false positive rate how trustable is the conclusion as a significant improvement.

Response 9: We thank the reviewer for their suggestions and comments. We have included the exact original value “0.228” to replace the term “somewhat high”. Patients with high circulating methylated CCDC181, GCM2 and ITPRIPL1 values will be evaluated their breast cancer risk after a 1-year follow-up period.

Point 10: In line 562, “Aberrant methylation of CCDC181 has been reported in lung and prostate.” Is it cancer or healthy tissues? If CCDC181 is found in these cancer types, then it means that it is not breast cancer specific biomarker as authors suggested.

Response 10: Although aberrant methylation of CCDC181 was found in lung and prostate cancer in previous studies and in endometrial cancer in Taiwanese cohort (Figure 2, shown as below), aberrant methylation of CCDC181 was a good cancer biomarker with high sensitivity (81.5%) in breast cancer. Combined analysis of high specificity breast cancer biomarker GCM2 and ITPRIPL1 can improve the sensitivity and specificity for breast cancer prediction. The description has been added to lines 648–653 of the discussion section.

  1. Uehiro, N.; Sato, F.; Pu, F.; Tanaka, S.; Kawashima, M.; Kawaguchi, K.; Sugimoto, M.; Saji, S.; Toi, M. Circulating cell-free DNA-based epigenetic assay can detect early breast cancer. Breast cancer research : BCR 2016, 18, 129, doi:10.1186/s13058-016-0788-z.
  2. Roos, L.; van Dongen, J.; Bell, C.G.; Burri, A.; Deloukas, P.; Boomsma, D.I.; Spector, T.D.; Bell, J.T. Integrative DNA methylome analysis of pan-cancer biomarkers in cancer discordant monozygotic twin-pairs. Clin Epigenetics 2016, 8, 7, doi:10.1186/s13148-016-0172-y.
  3. Schwarzenbach, H.; Milde-Langosch, K.; Steinbach, B.; Muller, V.; Pantel, K. Diagnostic potential of PTEN-targeting miR-214 in the blood of breast cancer patients. Breast Cancer Res Treat 2012, 134, 933-941, doi:10.1007/s10549-012-1988-6.
  4. Bacolod, M.D.; Huang, J.; Giardina, S.F.; Feinberg, P.B.; Mirza, A.H.; Swistel, A.; Soper, S.A.; Barany, F. Prediction of blood-based biomarkers and subsequent design of bisulfite PCR-LDR-qPCR assay for breast cancer detection. BMC cancer 2020, 20, 85, doi:10.1186/s12885-020-6574-4.
  5. Lin, N.; Liu, J.; Castle, J.; Wan, J.; Shendre, A.; Liu, Y.; Wang, C.; He, C. Genome-wide DNA methylation profiling in human breast tissue by illumina TruSeq methyl capture EPIC sequencing and infinium methylationEPIC beadchip microarray. Epigenetics 2020, 10.1080/15592294.2020.1827703, 1-16, doi:10.1080/15592294.2020.1827703.
  6. de Groot, J.S.; Moelans, C.B.; Elias, S.G.; Jo Fackler, M.; van Domselaar, R.; Suijkerbuijk, K.P.; Witkamp, A.J.; Sukumar, S.; van Diest, P.J.; van der Wall, E. DNA promoter hypermethylation in nipple fluid: a potential tool for early breast cancer detection. Oncotarget 2016, 7, 24778-24791, doi:10.18632/oncotarget.8352.
  7. Salas, L.A.; Lundgren, S.N.; Browne, E.P.; Punska, E.C.; Anderton, D.L.; Karagas, M.R.; Arcaro, K.F.; Christensen, B.C. Prediagnostic breast milk DNA methylation alterations in women who develop breast cancer. Hum Mol Genet 2020, 29, 662-673, doi:10.1093/hmg/ddz301.
  8. Yang, R.; Pfutze, K.; Zucknick, M.; Sutter, C.; Wappenschmidt, B.; Marme, F.; Qu, B.; Cuk, K.; Engel, C.; Schott, S., et al. DNA methylation array analyses identified breast cancer-associated HYAL2 methylation in peripheral blood. Int J Cancer 2015, 136, 1845-1855, doi:10.1002/ijc.29205.
  9. Toth, R.; Scherer, D.; Kelemen, L.E.; Risch, A.; Hazra, A.; Balavarca, Y.; Issa, J.J.; Moreno, V.; Eeles, R.A.; Ogino, S., et al. Genetic Variants in Epigenetic Pathways and Risks of Multiple Cancers in the GAME-ON Consortium. Cancer Epidemiol Biomarkers Prev 2017, 26, 816-825, doi:10.1158/1055-9965.Epi-16-0728.
  10. Tommasi, S.; Karm, D.L.; Wu, X.; Yen, Y.; Pfeifer, G.P. Methylation of homeobox genes is a frequent and early epigenetic event in breast cancer. Breast cancer research : BCR 2009, 11, R14, doi:10.1186/bcr2233.
  11. Herman, D.; Leakey, T.I.; Behrens, A.; Yao-Borengasser, A.; Cooney, C.A.; Jousheghany, F.; Phanavanh, B.; Siegel, E.R.; Safar, A.M.; Korourian, S., et al. CHST11 gene expression and DNA methylation in breast cancer. International journal of oncology 2015, 46, 1243-1251, doi:10.3892/ijo.2015.2828.
  12. Nordgard, S.H.; Johansen, F.E.; Alnaes, G.I.; Bucher, E.; Syvänen, A.C.; Naume, B.; Børresen-Dale, A.L.; Kristensen, V.N. Genome-wide analysis identifies 16q deletion associated with survival, molecular subtypes, mRNA expression, and germline haplotypes in breast cancer patients. Genes, chromosomes & cancer 2008, 47, 680-696, doi:10.1002/gcc.20569.
  13. Fernandez-Jimenez, N.; Sklias, A.; Ecsedi, S.; Cahais, V.; Degli-Esposti, D.; Jay, A.; Ancey, P.B.; Woo, H.D.; Hernandez-Vargas, H.; Herceg, Z. Lowly methylated region analysis identifies EBF1 as a potential epigenetic modifier in breast cancer. Epigenetics 2017, 12, 964-972, doi:10.1080/15592294.2017.1373919.
  14. Absmaier, M.; Napieralski, R.; Schuster, T.; Aubele, M.; Walch, A.; Magdolen, V.; Dorn, J.; Gross, E.; Harbeck, N.; Noske, A., et al. PITX2 DNA-methylation predicts response to anthracycline-based adjuvant chemotherapy in triple-negative breast cancer patients. International journal of oncology 2018, 52, 755-767, doi:10.3892/ijo.2018.4241.
  15. Fang, Z.; Yao, W.; Xiong, Y.; Zhang, J.; Liu, L.; Li, J.; Zhang, C.; Wan, J. Functional elucidation and methylation-mediated downregulation of ITGA5 gene in breast cancer cell line MDA-MB-468. J Cell Biochem 2010, 110, 1130-1141, doi:10.1002/jcb.22626.
  16. Liu, K.; Dong, F.; Gao, H.; Guo, Y.; Li, H.; Yang, F.; Zhao, P.; Dai, Y.; Wang, J.; Zhou, W., et al. Promoter hypermethylation of the CFTR gene as a novel diagnostic and prognostic marker of breast cancer. Cell biology international 2020, 44, 603-609, doi:10.1002/cbin.11260.
  17. Xiao, B.; Chen, L.; Ke, Y.; Hang, J.; Cao, L.; Zhang, R.; Zhang, W.; Liao, Y.; Gao, Y.; Chen, J., et al. Identification of methylation sites and signature genes with prognostic value for luminal breast cancer. BMC cancer 2018, 18, 405, doi:10.1186/s12885-018-4314-9.
  18. Lindqvist, B.M.; Wingren, S.; Motlagh, P.B.; Nilsson, T.K. Whole genome DNA methylation signature of HER2-positive breast cancer. Epigenetics 2014, 9, 1149-1162, doi:10.4161/epi.29632.
  19. Daino, K.; Nishimura, M.; Imaoka, T.; Takabatake, M.; Morioka, T.; Nishimura, Y.; Shimada, Y.; Kakinuma, S. Epigenetic dysregulation of key developmental genes in radiation-induced rat mammary carcinomas. Int J Cancer 2018, 143, 343-354, doi:10.1002/ijc.31309.
  20. Zhang, C.; Zhao, H.; Li, J.; Liu, H.; Wang, F.; Wei, Y.; Su, J.; Zhang, D.; Liu, T.; Zhang, Y. The identification of specific methylation patterns across different cancers. PloS one 2015, 10, e0120361, doi:10.1371/journal.pone.0120361.
  21. Faryna, M.; Konermann, C.; Aulmann, S.; Bermejo, J.L.; Brugger, M.; Diederichs, S.; Rom, J.; Weichenhan, D.; Claus, R.; Rehli, M., et al. Genome-wide methylation screen in low-grade breast cancer identifies novel epigenetically altered genes as potential biomarkers for tumor diagnosis. FASEB journal : official publication of the Federation of American Societies for Experimental Biology 2012, 26, 4937-4950, doi:10.1096/fj.12-209502.
  22. Strelnikov, V.V.; Kuznetsova, E.B.; Tanas, A.S.; Rudenko, V.V.; Kalinkin, A.I.; Poddubskaya, E.V.; Kekeeva, T.V.; Chesnokova, G.G.; Trotsenko, I.D.; Larin, S.S., et al. Abnormal promoter DNA hypermethylation of the integrin, nidogen, and dystroglycan genes in breast cancer. Scientific reports 2021, 11, 2264, doi:10.1038/s41598-021-81851-y.
  23. Jeschke, J.; Van Neste, L.; Glöckner, S.C.; Dhir, M.; Calmon, M.F.; Deregowski, V.; Van Criekinge, W.; Vlassenbroeck, I.; Koch, A.; Chan, T.A., et al. Biomarkers for detection and prognosis of breast cancer identified by a functional hypermethylome screen. Epigenetics 2012, 7, 701-709, doi:10.4161/epi.20445.
  24. Ennour-Idrissi, K.; Dragic, D.; Issa, E.; Michaud, A.; Chang, S.L.; Provencher, L.; Durocher, F.; Diorio, C. DNA Methylation and Breast Cancer Risk: An Epigenome-Wide Study of Normal Breast Tissue and Blood. Cancers 2020, 12, doi:10.3390/cancers12113088.
  25. Zhang, Y.; Yang, R.; Burwinkel, B.; Breitling, L.P.; Holleczek, B.; Schöttker, B.; Brenner, H. F2RL3 methylation in blood DNA is a strong predictor of mortality. International journal of epidemiology 2014, 43, 1215-1225, doi:10.1093/ije/dyu006.
  26. Lee, S.E.; Kim, S.J.; Yoon, H.J.; Yu, S.Y.; Yang, H.; Jeong, S.I.; Hwang, S.Y.; Park, C.S.; Park, Y.S. Genome-wide profiling in melatonin-exposed human breast cancer cell lines identifies differentially methylated genes involved in the anticancer effect of melatonin. Journal of pineal research 2013, 54, 80-88, doi:10.1111/j.1600-079X.2012.01027.x.
  27. Halvorsen, A.R.; Helland, A.; Fleischer, T.; Haug, K.M.; Grenaker Alnaes, G.I.; Nebdal, D.; Syljuåsen, R.G.; Touleimat, N.; Busato, F.; Tost, J., et al. Differential DNA methylation analysis of breast cancer reveals the impact of immune signaling in radiation therapy. Int J Cancer 2014, 135, 2085-2095, doi:10.1002/ijc.28862.
  28. Kanazawa, T.; Misawa, K.; Fukushima, H.; Misawa, Y.; Sato, Y.; Maruta, M.; Imayoshi, S.; Kusaka, G.; Kawabata, K.; Mineta, H., et al. Epigenetic inactivation of galanin receptors in salivary duct carcinoma of the parotid gland: Potential utility as biomarkers for prognosis. Oncology letters 2018, 15, 9043-9050, doi:10.3892/ol.2018.8525.
  29. Khakpour, G.; Noruzinia, M.; Izadi, P.; Karami, F.; Ahmadvand, M.; Heshmat, R.; Amoli, M.M.; Tavakkoly-Bazzaz, J. Methylomics of breast cancer: Seeking epimarkers in peripheral blood of young subjects. Tumour Biol 2017, 39, 1010428317695040, doi:10.1177/1010428317695040.
  30. Lo Nigro, C.; Monteverde, M.; Lee, S.; Lattanzio, L.; Vivenza, D.; Comino, A.; Syed, N.; McHugh, A.; Wang, H.; Proby, C., et al. NT5E CpG island methylation is a favourable breast cancer biomarker. Br J Cancer 2012, 107, 75-83, doi:10.1038/bjc.2012.212.
  31. Fan, Y.; He, L.; Wang, Y.; Fu, S.; Han, Y.; Fan, J.; Wen, Q. CLIP4 Shows Putative Tumor Suppressor Characteristics in Breast Cancer: An Integrated Analysis. Front Mol Biosci 2020, 7, 616190, doi:10.3389/fmolb.2020.616190.
  32. Zhu, Q.; Wang, J.; Zhang, Q.; Wang, F.; Fang, L.; Song, B.; Xie, C.; Liu, J. Methylationdriven genes PMPCAP1, SOWAHC and ZNF454 as potential prognostic biomarkers in lung squamous cell carcinoma. Mol Med Rep 2020, 21, 1285-1295, doi:10.3892/mmr.2020.10933.
  33. Simonova, O.A.; Kuznetsova, E.B.; Tanas, A.S.; Rudenko, V.V.; Poddubskaya, E.V.; Kekeeva, T.V.; Trotsenko, I.D.; Larin, S.S.; Kutsev, S.I.; Zaletaev, D.V., et al. Abnormal Hypermethylation of CpG Dinucleotides in Promoter Regions of Matrix Metalloproteinases Genes in Breast Cancer and Its Relation to Epigenomic Subtypes and HER2 Overexpression. Biomedicines 2020, 8, doi:10.3390/biomedicines8050116.
  34. Mao, X.H.; Ye, Q.; Zhang, G.B.; Jiang, J.Y.; Zhao, H.Y.; Shao, Y.F.; Ye, Z.Q.; Xuan, Z.X.; Huang, P. Identification of differentially methylated genes as diagnostic and prognostic biomarkers of breast cancer. World J Surg Oncol 2021, 19, 29, doi:10.1186/s12957-021-02124-6.
  35. Liang, Z.Z.; Zhu, R.M.; Li, Y.L.; Jiang, H.M.; Li, R.B.; Tang, L.Y.; Wang, Q.; Ren, Z.F. Differential epigenetic and transcriptional profile in MCF-7 breast cancer cells exposed to cadmium. Chemosphere 2020, 261, 128148, doi:10.1016/j.chemosphere.2020.128148.
  36. Li, S.Y.; Wu, H.C.; Mai, H.F.; Zhen, J.X.; Li, G.S.; Chen, S.J. Microarray-based analysis of whole-genome DNA methylation profiling in early detection of breast cancer. J Cell Biochem 2019, 120, 658-670, doi:10.1002/jcb.27423.
  37. Dietrich, D.; Lesche, R.; Tetzner, R.; Krispin, M.; Dietrich, J.; Haedicke, W.; Schuster, M.; Kristiansen, G. Analysis of DNA methylation of multiple genes in microdissected cells from formalin-fixed and paraffin-embedded tissues. The journal of histochemistry and cytochemistry : official journal of the Histochemistry Society 2009, 57, 477-489, doi:10.1369/jhc.2009.953026.
  38. Flanagan, J.M.; Brook, M.N.; Orr, N.; Tomczyk, K.; Coulson, P.; Fletcher, O.; Jones, M.E.; Schoemaker, M.J.; Ashworth, A.; Swerdlow, A., et al. Temporal stability and determinants of white blood cell DNA methylation in the breakthrough generations study. Cancer Epidemiol Biomarkers Prev 2015, 24, 221-229, doi:10.1158/1055-9965.Epi-14-0767.
  39. Dumitrescu, R.G. Early Epigenetic Markers for Precision Medicine. Methods in molecular biology (Clifton, N.J.) 2018, 1856, 3-17, doi:10.1007/978-1-4939-8751-1_1.
  40. Dookeran, K.A.; Zhang, W.; Stayner, L.; Argos, M. Associations of two-pore domain potassium channels and triple negative breast cancer subtype in The Cancer Genome Atlas: systematic evaluation of gene expression and methylation. BMC research notes 2017, 10, 475, doi:10.1186/s13104-017-2777-4.
  41. Zhang, S.; Wang, Y.; Gu, Y.; Zhu, J.; Ci, C.; Guo, Z.; Chen, C.; Wei, Y.; Lv, W.; Liu, H., et al. Specific breast cancer prognosis-subtype distinctions based on DNA methylation patterns. Molecular oncology 2018, 12, 1047-1060, doi:10.1002/1878-0261.12309.
  42. Wilson, L.E.; Harlid, S.; Xu, Z.; Sandler, D.P.; Taylor, J.A. An epigenome-wide study of body mass index and DNA methylation in blood using participants from the Sister Study cohort. International journal of obesity (2005) 2017, 41, 194-199, doi:10.1038/ijo.2016.184.
  43. Tessema, M.; Yu, Y.Y.; Stidley, C.A.; Machida, E.O.; Schuebel, K.E.; Baylin, S.B.; Belinsky, S.A. Concomitant promoter methylation of multiple genes in lung adenocarcinomas from current, former and never smokers. Carcinogenesis 2009, 30, 1132-1138, doi:10.1093/carcin/bgp114.
  44. Pangeni, R.P.; Channathodiyil, P.; Huen, D.S.; Eagles, L.W.; Johal, B.K.; Pasha, D.; Hadjistephanou, N.; Nevell, O.; Davies, C.L.; Adewumi, A.I., et al. The GALNT9, BNC1 and CCDC8 genes are frequently epigenetically dysregulated in breast tumours that metastasise to the brain. Clin Epigenetics 2015, 7, 57, doi:10.1186/s13148-015-0089-x.
  45. Klassen, L.M.B.; Chequin, A.; Manica, G.C.M.; Biembengut, I.V.; Toledo, M.B.; Baura, V.A.; de, O.P.F.; Ramos, E.A.S.; Costa, F.F.; de Souza, E.M., et al. MMP9 gene expression regulation by intragenic epigenetic modifications in breast cancer. Gene 2018, 642, 461-466, doi:10.1016/j.gene.2017.11.054.
  46. Kresovich, J.K.; Gann, P.H.; Erdal, S.; Chen, H.Y.; Argos, M.; Rauscher, G.H. Candidate gene DNA methylation associations with breast cancer characteristics and tumor progression. Epigenomics 2018, 10, 367-378, doi:10.2217/epi-2017-0119.
  47. Ishihara, H.; Yamashita, S.; Fujii, S.; Tanabe, K.; Mukai, H.; Ushijima, T. DNA methylation marker to estimate the breast cancer cell fraction in DNA samples. Medical oncology (Northwood, London, England) 2018, 35, 147, doi:10.1007/s12032-018-1207-3.

Reviewer 3 Report

Sheng-Chao Wang et. al used genome-wide DNA methylation array integrated with TCGA BC patient’s data to develop a new BC prediction biomarker including with CCDC181, GCM2 and ITPRIPL1 genes. This new prediction model has high sensitivity and AUC, which is promising for the BC biomarkers clinically. However, there are some concerns need to be addressed:

Major:

  1. Please address the logic to only focus on those hypermethylation in BC patients and close to 0 in normal tissue. Why discard those hypomethylation in BC and hypermethylation in normal tissue?
  2. How many genes overlapped between 1612 genes and 2865 genes from TCGA? How to come down to 7 candidate genes from thousand genes?
  3. For result 3.2, the description of the seven candidate genes is kind of messy. Especially for which genes have uniform aberrant DNA methylation level in which subtypes of breast cancer. The authors may need to rewrite this paragraph to emphasize that the final three genes.
  4. When author mentioned that the methylation levels of CCDC181, GCM2 …, does author means the mean gene body DNA methylation level for these genes? How many CpGs measured for each gene?
  5. For figure S2, I suggest using absolute DNA methylation level as Y-axis. The fold change of DNA methylation level does not mean anything. For example, you can have a 0.01 vs 0.2 which has 200 increase fold changes, but still keep very low DNA methylation level.
  6. In result 3.3, I assume that the specificity mentioned by author means that specific high DNA methylation level in breast tumor/plasma than in normal tissue/plasma. However, this specificity can also mean that compare to other types of solid tumor, these biomarkers are specific high DNA methylation level in breast cancer, not other cancer. The author may clarify these or add more analysis include with other cancer type data.

The average beta value could be misleading when the data has large variance. So it is important to use boxplots to show these biomarkers have uniform low DNA methylation level in other cancer types.

  1. In the whole text, there are several times that the author included new cohort of patients, please clarify the subtypes of these breast cancer patients. Different subtypes breast cancer has distinct DNA methylation patterns.
  2. Figure 2a, b, color bar scale is too broad, which will cover many variations in real data. Please use more refined color scale.
  3. Figure 2c, d. I am confused about the normalized methylation level to ACTB gene. It is not right. The ACTB gene expression can be used to normalize the gene expression level. There is no data showing that ACTB gene body/promoter DNA methylation level is maintained in the same level in all tissues, cell types, conditions/diseases. Unless the author can provide these evidences. The y-axis should be the absolute DNA methylation values. The same for Figure 3 and 4.
  4. The author claim it is cell free DNA methylation from plasma, they should provide bioanalyzer results for the fragment length distribution to prove they are using the cell free DNA not genome DNA.
  5. Again, Fig 5, please use absolute DNA methylation level in Y-axis.
  6. Please provide the age information for patients.

Author Response

Response to Reviewer 3 comments:

Point 1: Please address the logic to only focus on those hypermethylation in BC patients and close to 0 in normal tissue. Why discard those hypomethylation in BC and hypermethylation in normal tissue?

Response 1:

We indeed selected several hypomethylated genes, such as TSTD1, for further analysis in breast cancer tissues. A decrease in the methylation level of TSTD1 was observed in breast tumor tissue compared to adjacent normal tissue (shown as below). However, the assessment of circulating hypomethylated TSTD1 was not able to distinguish the plasma from breast cancer patients or healthy subjects in our preliminary study, as the CCDC181, GCM2 and ITPRIPL1 genes could. The analysis has not been completed. It is doubt whether circulating hypomethylated ccfDNA of genes could work well as early prediction biomarkers for breast cancer according to current data. Therefore, we focused on the identification of promotor hypermethylation in genes in breast cancer patients from a Western cohort from the TCGA database and a Taiwan cohort in this manuscript.

We thank the reviewer for their suggestions. We have addressed the logical and rational reasons as to why we focused on the hypermethylated genes for breast cancer early prediction in line 612-617 of the Discussion section.

Point 2: How many genes overlapped between 1612 genes and 2865 genes from TCGA? How to come down to 7 candidate genes from thousand genes?

Response 2: A total of 1190 genes were found to be overlapped between 1612 genes in the Taiwanese cohort and 2865 genes in the TCGA cohort. This study focused on the top 200 genes with the highest methylation levels in breast tumors from TCGA. In total, 160 highly methylated genes were found when the genes identified from the Taiwanese and TCGA cohorts were combined (Table S2). A literature review was conducted and the following previously-reported aberrantly methylated genes were excluded: DPF1 [1], DNM3 [2], TLX1 [3], CHST11 [4], CHST3 [5], ALX1 [6], HNF1B [3], EBF1 [7], PITX2 [8], ITGA5 [9], COL11A2 [10], CFTR [11], NR5A2 [12], PRKCB [12], SOSTDC1 [13], DBX1 [14], LHX8 [15], HCK [16], NXPH1 [17], NID2 [18], TAC1 [19], TFAP2B [20], F2RL3 [21], POU4F2 [22], H2AFY [23], GALR1 [24], FOXD3 [25], NT5E [26], CLIP4 [27], ZNF454 [28], TIMP2 [29], GNG4 [30], MIR129-2 [30], TXNRD1 [31], CPXM1 [30], PRDM14 [30], HOXA4 [32], SLITRK1 [33], AHRR [34], NPTX2 [35], KCNK9 [36], C9orf122 [37], CRHR2 [38], BOLL [39], CCDC8 [40], MMP9 [41], C12orf68 [37]. The remaining candidate genes were then studied through sequencing and gene-specific primers/probe design to determine their methylation-specific PCR reactions. The results for seven candidate genes—CCDC181, GCM2, ITPRIPL1, ENPP2, LOC643719, ZNF177, and ADCY4—were successfully validated by sequencing and stable detection in ccfDNA from the plasma of Taiwanese patients with breast cancer. The seven candidate genes, CCDC181, GCM2, ITPRIPL1, ENPP2, LOC643719, ZNF177, ADCY4, were found to be hypermethylated in both the Taiwanese and Western cohorts (Figure 1). RASSF1 has been reported to be hypermethylated in breast cancer and showed a similar pattern to the reference control in breast cancer patients from both the Taiwan and Western cohorts [42]. Therefore, CCDC181, GCM2, ITPRIPL1, ENPP2, LOC643719, ZNF177, ADCY4, and RASSF1 were analyzed in the following assays.

Point 3: For result 3.2, the description of the seven candidate genes is kind of messy. Especially for which genes have uniform aberrant DNA methylation level in which subtypes of breast cancer. The authors may need to rewrite this paragraph to emphasize that the final three genes.

Response 3: We thank the reviewer for their suggestions. The description has been modified in the line 374-379 and of result 3.2 and 3.3.

For result 3.2: Hypermethylation of CCDC181 and GCM2 could be detected in ductal carcinoma in situ (DCIS) tumors of breast cancer patients (Table 1).  Hypermethylation of CCDC181, GCM2, and ITPRIPL1 was found in more than 80% of patients with lower tumor histologic grades, and in 70% of early-stage breast cancer among all subtypes of breast cancer patients (Table 1). There was no significant difference between hypermethylation of CCDC181, GCM2, and ITPRIPL1 with expression of ER and PR. However, hypermethylation of GCM2 and ITPRIPL1 was significantly associated with HER2 positivity (Table 1, P = 0.031 and 0.011).

For result 3.3: Hypermethylation of CCDC181, GCM2 and ITPRIPL1 was observed in different subgroups of breast cancer patients, including in individuals of different races and in various menopause states as well as with different tumor stages and histological types (Table S4).

Point 4: When author mentioned that the methylation levels of CCDC181, GCM2 …, does author means the mean gene body DNA methylation level for these genes? How many CpGs measured for each gene?

Response 4: We thank the reviewer for their suggestions. The primers/probes for candidate genes were designed on their methylated promotor regions, especially on the identified differential regions between normal and tumor tissues. A total of 6 CpGs were designed on the CCDC181 gene, 7 CpGs were designed on the GCM2 gene, and 5 CpGs were designed on the ITPRIPL1 gene. According to the sequencing results, only when all CpG sites are methylated can a successful PCR reaction occur.

The description has been added to Section 2.8.

Gene

primer

5’→3’sequences

Application

Size

(bp)

Tm ( )

BACTIN

Forward

Reverse

Probe

TGGTGATGGAGGAGGTTTAGTAAGT

AACCAATAAAACCTACTCCTCCCTTAA

ACCACCACCCAACACACAATAACAAACACA

MSP-M

132

60

CCDC181

Forward

Reverse

Probe

TTTTATTGGTTTTTCGTAAGTATCG

CATAACAACAACGTACCTCTACGTC

TCGGGAGGGGTCGGTGGTTTGAG

MSP-M

143

60

GCM2

Forward

Reverse

Probe

GAGATAGGGCGGAGTTTTTC

CTTAACCGCGATACTAAACGTT

TCCACCCGAACGACAACATCGACC

MSP-M

105

60

ITPRIPL1

Forward

Reverse

Probe

GAGTGTAGTTGATAGTAGGTACGGC

GTAAATTTACTAAAAAAATAAAAAAACCGT

CACACTCTCCGCTACTCGACCTCCCTA

MSP-M

106

60

Point 5: For figure S2, I suggest using absolute DNA methylation level as Y-axis. The fold change of DNA methylation level does not mean anything. For example, you can have a 0.01 vs 0.2 which has 200 increase fold changes, but still keep very low DNA methylation level.

Response 5: We thankful the reviewer for the suggestions and comments. The Figure S2 has been replotted.

Point 6: In result 3.3, I assume that the specificity mentioned by author means that specific high DNA methylation level in breast tumor/plasma than in normal tissue/plasma. However, this specificity can also mean that compare to other types of solid tumor, these biomarkers are specific high DNA methylation level in breast cancer, not other cancer. The author may clarify these or add more analysis include with other cancer type data. So it is important to use boxplots to show these biomarkers have uniform low DNA methylation level in other cancer types.

Response 6: We thank the reviewer for their suggestions and comments. The figure shows the DNA methylation levels of the CCDC181, GCM2, and ITPRIPL1 genes in different Taiwanese cancer types using boxplots (Figure 2).

Point 7: In the whole text, there are several times that the author included new cohort of patients, please clarify the subtypes of these breast cancer patients. Different subtypes breast cancer has distinct DNA methylation patterns.

Response 7: We thank the reviewer for their suggestions and comments. We have addressed the subtypes of breast cancer patients in different cohort in Table S1.

Point 8: Figure 3a, 3b, color bar scale is too broad, which will cover many variations in real data. Please use more refined color scale.

Response 8: We thank the reviewer for their suggestions and comments. We have replotted the heatmap in the Figure 3a, 3b using heatmapper software. A gradient scale heatmap (100 color categories) was used to visualize the DNA methylation level from low to high (yellow to blue) [43]. The description has been added to Section 2.7.

Point 9: Figure 3c,3d. I am confused about the normalized methylation level to ACTB gene. The ACTB gene expression can be used to normalize the gene expression level. There is no data showing that ACTB gene body/promoter DNA methylation level is maintained in the same level in all tissues, cell types, conditions/diseases. Unless the author can provide these evidences. The y-axis should be the absolute DNA methylation values. The same for Figure 3 and 4.

Response 9: We apologized that we didn’t describe clear. The beta-actin (ACTB) gene was used as methylation-independent DNA control. The primer design for ACTB PCR assay (without CpG dinucleotides) is methylation-independent acting as DNA input control, has been widely used as an internal control for DNA methylation analysis [44-46].

Table S1. List of primer sequences and conditions used in the present study.

Gene

primer

5’→3’sequences

Application

Size

(bp)

Tm ( )

BACTIN

Forward

Reverse

Probe

TGGTGATGGAGGAGGTTTAGTAAGT

AACCAATAAAACCTACTCCTCCCTTAA

ACCACCACCCAACACACAATAACAAACACA

MSP

(Input DNA control)

132

60

Previous studies reported that ACTB was used for quantitative measurement of cell-free plasma DNA when analyzing DNA methylation [47]; ACTB was applied as an internal reference gene for DNA Methylation Marker Panel for Plasma-Based Discrimination between Patients with Malignant and Nonmalignant Lung Disease [48]; ACTB was applied as quality control gene for reliable DNA methylation analyses in liquid biopsies [49]. The description of primer design for ACTB has been added in the line 259-266.

Point 10: The author claim it is cell free DNA methylation from plasma, they should provide bioanalyzer results for the fragment length distribution to prove they are using the cell free DNA not genome DNA.

Response 10: We thank the reviewer for their suggestions and comments. The efficiency of short DNA recovery is shown below. According to the manufacturer’s instructions for the MagMAX™ cell-free DNA isolation kit (ThermoFisher Scientific, TX, USA, A29319), only DNA near 170 bases, which is normally considered to be derived from ccfDNA, was present after extraction. Approximately 100% of ccfDNA ranging from 100 to 750 bp was recovered.

Previous reports have shown ccfDNA samples with clear fragment size peaks between 140 and 200 bp [33]. The MagMAX Cell-free DNA Isolation kit provided the highest yield and low molecular weight fractions [34]. The description has been added to lines 194–196.

Point 11: Again, Fig 5, please use absolute DNA methylation level in Y-axis.

Response 11: We thank the reviewer for their suggestions and comments. Because the absolute DNA methylation level in stage 0 and stage I is much less than in stage II and stage III, it is difficult to conclude that the aberrant circulating hypermethylated CCDC181, GCM2, and ITPRIPL1 levels were dramatically decreased after surgery in the earlier disease stages (0 and I) without lymph nodes reginal metastasis. The results of circulating methylated CCDC181 was shown below. Therefore, the fold change of DNA methylation was used to reveal the loss of aberrant circulating hypermethylated CCDC181, GCM2, and ITPRIPL1 after surgery in this figure.

Point 12: Please provide the age information for patients.

Response 12: We thank the reviewer for their suggestions and comments. We have added the age information for patients to table 1.

  1. Toth, R.; Scherer, D.; Kelemen, L.E.; Risch, A.; Hazra, A.; Balavarca, Y.; Issa, J.J.; Moreno, V.; Eeles, R.A.; Ogino, S., et al. Genetic Variants in Epigenetic Pathways and Risks of Multiple Cancers in the GAME-ON Consortium. Cancer Epidemiol Biomarkers Prev 2017, 26, 816-825, doi:10.1158/1055-9965.Epi-16-0728.
  2. Uehiro, N.; Sato, F.; Pu, F.; Tanaka, S.; Kawashima, M.; Kawaguchi, K.; Sugimoto, M.; Saji, S.; Toi, M. Circulating cell-free DNA-based epigenetic assay can detect early breast cancer. Breast cancer research : BCR 2016, 18, 129, doi:10.1186/s13058-016-0788-z.
  3. Tommasi, S.; Karm, D.L.; Wu, X.; Yen, Y.; Pfeifer, G.P. Methylation of homeobox genes is a frequent and early epigenetic event in breast cancer. Breast cancer research : BCR 2009, 11, R14, doi:10.1186/bcr2233.
  4. Herman, D.; Leakey, T.I.; Behrens, A.; Yao-Borengasser, A.; Cooney, C.A.; Jousheghany, F.; Phanavanh, B.; Siegel, E.R.; Safar, A.M.; Korourian, S., et al. CHST11 gene expression and DNA methylation in breast cancer. International journal of oncology 2015, 46, 1243-1251, doi:10.3892/ijo.2015.2828.
  5. Nordgard, S.H.; Johansen, F.E.; Alnaes, G.I.; Bucher, E.; Syvänen, A.C.; Naume, B.; Børresen-Dale, A.L.; Kristensen, V.N. Genome-wide analysis identifies 16q deletion associated with survival, molecular subtypes, mRNA expression, and germline haplotypes in breast cancer patients. Genes, chromosomes & cancer 2008, 47, 680-696, doi:10.1002/gcc.20569.
  6. de Groot, J.S.; Moelans, C.B.; Elias, S.G.; Jo Fackler, M.; van Domselaar, R.; Suijkerbuijk, K.P.; Witkamp, A.J.; Sukumar, S.; van Diest, P.J.; van der Wall, E. DNA promoter hypermethylation in nipple fluid: a potential tool for early breast cancer detection. Oncotarget 2016, 7, 24778-24791, doi:10.18632/oncotarget.8352.
  7. Fernandez-Jimenez, N.; Sklias, A.; Ecsedi, S.; Cahais, V.; Degli-Esposti, D.; Jay, A.; Ancey, P.B.; Woo, H.D.; Hernandez-Vargas, H.; Herceg, Z. Lowly methylated region analysis identifies EBF1 as a potential epigenetic modifier in breast cancer. Epigenetics 2017, 12, 964-972, doi:10.1080/15592294.2017.1373919.
  8. Absmaier, M.; Napieralski, R.; Schuster, T.; Aubele, M.; Walch, A.; Magdolen, V.; Dorn, J.; Gross, E.; Harbeck, N.; Noske, A., et al. PITX2 DNA-methylation predicts response to anthracycline-based adjuvant chemotherapy in triple-negative breast cancer patients. International journal of oncology 2018, 52, 755-767, doi:10.3892/ijo.2018.4241.
  9. Fang, Z.; Yao, W.; Xiong, Y.; Zhang, J.; Liu, L.; Li, J.; Zhang, C.; Wan, J. Functional elucidation and methylation-mediated downregulation of ITGA5 gene in breast cancer cell line MDA-MB-468. J Cell Biochem 2010, 110, 1130-1141, doi:10.1002/jcb.22626.
  10. Roos, L.; van Dongen, J.; Bell, C.G.; Burri, A.; Deloukas, P.; Boomsma, D.I.; Spector, T.D.; Bell, J.T. Integrative DNA methylome analysis of pan-cancer biomarkers in cancer discordant monozygotic twin-pairs. Clin Epigenetics 2016, 8, 7, doi:10.1186/s13148-016-0172-y.
  11. Liu, K.; Dong, F.; Gao, H.; Guo, Y.; Li, H.; Yang, F.; Zhao, P.; Dai, Y.; Wang, J.; Zhou, W., et al. Promoter hypermethylation of the CFTR gene as a novel diagnostic and prognostic marker of breast cancer. Cell biology international 2020, 44, 603-609, doi:10.1002/cbin.11260.
  12. Bacolod, M.D.; Huang, J.; Giardina, S.F.; Feinberg, P.B.; Mirza, A.H.; Swistel, A.; Soper, S.A.; Barany, F. Prediction of blood-based biomarkers and subsequent design of bisulfite PCR-LDR-qPCR assay for breast cancer detection. BMC cancer 2020, 20, 85, doi:10.1186/s12885-020-6574-4.
  13. Xiao, B.; Chen, L.; Ke, Y.; Hang, J.; Cao, L.; Zhang, R.; Zhang, W.; Liao, Y.; Gao, Y.; Chen, J., et al. Identification of methylation sites and signature genes with prognostic value for luminal breast cancer. BMC cancer 2018, 18, 405, doi:10.1186/s12885-018-4314-9.
  14. Lindqvist, B.M.; Wingren, S.; Motlagh, P.B.; Nilsson, T.K. Whole genome DNA methylation signature of HER2-positive breast cancer. Epigenetics 2014, 9, 1149-1162, doi:10.4161/epi.29632.
  15. Daino, K.; Nishimura, M.; Imaoka, T.; Takabatake, M.; Morioka, T.; Nishimura, Y.; Shimada, Y.; Kakinuma, S. Epigenetic dysregulation of key developmental genes in radiation-induced rat mammary carcinomas. Int J Cancer 2018, 143, 343-354, doi:10.1002/ijc.31309.
  16. Zhang, C.; Zhao, H.; Li, J.; Liu, H.; Wang, F.; Wei, Y.; Su, J.; Zhang, D.; Liu, T.; Zhang, Y. The identification of specific methylation patterns across different cancers. PloS one 2015, 10, e0120361, doi:10.1371/journal.pone.0120361.
  17. Faryna, M.; Konermann, C.; Aulmann, S.; Bermejo, J.L.; Brugger, M.; Diederichs, S.; Rom, J.; Weichenhan, D.; Claus, R.; Rehli, M., et al. Genome-wide methylation screen in low-grade breast cancer identifies novel epigenetically altered genes as potential biomarkers for tumor diagnosis. FASEB journal : official publication of the Federation of American Societies for Experimental Biology 2012, 26, 4937-4950, doi:10.1096/fj.12-209502.
  18. Strelnikov, V.V.; Kuznetsova, E.B.; Tanas, A.S.; Rudenko, V.V.; Kalinkin, A.I.; Poddubskaya, E.V.; Kekeeva, T.V.; Chesnokova, G.G.; Trotsenko, I.D.; Larin, S.S., et al. Abnormal promoter DNA hypermethylation of the integrin, nidogen, and dystroglycan genes in breast cancer. Scientific reports 2021, 11, 2264, doi:10.1038/s41598-021-81851-y.
  19. Jeschke, J.; Van Neste, L.; Glöckner, S.C.; Dhir, M.; Calmon, M.F.; Deregowski, V.; Van Criekinge, W.; Vlassenbroeck, I.; Koch, A.; Chan, T.A., et al. Biomarkers for detection and prognosis of breast cancer identified by a functional hypermethylome screen. Epigenetics 2012, 7, 701-709, doi:10.4161/epi.20445.
  20. Ennour-Idrissi, K.; Dragic, D.; Issa, E.; Michaud, A.; Chang, S.L.; Provencher, L.; Durocher, F.; Diorio, C. DNA Methylation and Breast Cancer Risk: An Epigenome-Wide Study of Normal Breast Tissue and Blood. Cancers 2020, 12, doi:10.3390/cancers12113088.
  21. Zhang, Y.; Yang, R.; Burwinkel, B.; Breitling, L.P.; Holleczek, B.; Schöttker, B.; Brenner, H. F2RL3 methylation in blood DNA is a strong predictor of mortality. International journal of epidemiology 2014, 43, 1215-1225, doi:10.1093/ije/dyu006.
  22. Lee, S.E.; Kim, S.J.; Yoon, H.J.; Yu, S.Y.; Yang, H.; Jeong, S.I.; Hwang, S.Y.; Park, C.S.; Park, Y.S. Genome-wide profiling in melatonin-exposed human breast cancer cell lines identifies differentially methylated genes involved in the anticancer effect of melatonin. Journal of pineal research 2013, 54, 80-88, doi:10.1111/j.1600-079X.2012.01027.x.
  23. Halvorsen, A.R.; Helland, A.; Fleischer, T.; Haug, K.M.; Grenaker Alnaes, G.I.; Nebdal, D.; Syljuåsen, R.G.; Touleimat, N.; Busato, F.; Tost, J., et al. Differential DNA methylation analysis of breast cancer reveals the impact of immune signaling in radiation therapy. Int J Cancer 2014, 135, 2085-2095, doi:10.1002/ijc.28862.
  24. Kanazawa, T.; Misawa, K.; Fukushima, H.; Misawa, Y.; Sato, Y.; Maruta, M.; Imayoshi, S.; Kusaka, G.; Kawabata, K.; Mineta, H., et al. Epigenetic inactivation of galanin receptors in salivary duct carcinoma of the parotid gland: Potential utility as biomarkers for prognosis. Oncology letters 2018, 15, 9043-9050, doi:10.3892/ol.2018.8525.
  25. Khakpour, G.; Noruzinia, M.; Izadi, P.; Karami, F.; Ahmadvand, M.; Heshmat, R.; Amoli, M.M.; Tavakkoly-Bazzaz, J. Methylomics of breast cancer: Seeking epimarkers in peripheral blood of young subjects. Tumour Biol 2017, 39, 1010428317695040, doi:10.1177/1010428317695040.
  26. Lo Nigro, C.; Monteverde, M.; Lee, S.; Lattanzio, L.; Vivenza, D.; Comino, A.; Syed, N.; McHugh, A.; Wang, H.; Proby, C., et al. NT5E CpG island methylation is a favourable breast cancer biomarker. Br J Cancer 2012, 107, 75-83, doi:10.1038/bjc.2012.212.
  27. Fan, Y.; He, L.; Wang, Y.; Fu, S.; Han, Y.; Fan, J.; Wen, Q. CLIP4 Shows Putative Tumor Suppressor Characteristics in Breast Cancer: An Integrated Analysis. Front Mol Biosci 2020, 7, 616190, doi:10.3389/fmolb.2020.616190.
  28. Zhu, Q.; Wang, J.; Zhang, Q.; Wang, F.; Fang, L.; Song, B.; Xie, C.; Liu, J. Methylationdriven genes PMPCAP1, SOWAHC and ZNF454 as potential prognostic biomarkers in lung squamous cell carcinoma. Mol Med Rep 2020, 21, 1285-1295, doi:10.3892/mmr.2020.10933.
  29. Simonova, O.A.; Kuznetsova, E.B.; Tanas, A.S.; Rudenko, V.V.; Poddubskaya, E.V.; Kekeeva, T.V.; Trotsenko, I.D.; Larin, S.S.; Kutsev, S.I.; Zaletaev, D.V., et al. Abnormal Hypermethylation of CpG Dinucleotides in Promoter Regions of Matrix Metalloproteinases Genes in Breast Cancer and Its Relation to Epigenomic Subtypes and HER2 Overexpression. Biomedicines 2020, 8, doi:10.3390/biomedicines8050116.
  30. Mao, X.H.; Ye, Q.; Zhang, G.B.; Jiang, J.Y.; Zhao, H.Y.; Shao, Y.F.; Ye, Z.Q.; Xuan, Z.X.; Huang, P. Identification of differentially methylated genes as diagnostic and prognostic biomarkers of breast cancer. World J Surg Oncol 2021, 19, 29, doi:10.1186/s12957-021-02124-6.
  31. Liang, Z.Z.; Zhu, R.M.; Li, Y.L.; Jiang, H.M.; Li, R.B.; Tang, L.Y.; Wang, Q.; Ren, Z.F. Differential epigenetic and transcriptional profile in MCF-7 breast cancer cells exposed to cadmium. Chemosphere 2020, 261, 128148, doi:10.1016/j.chemosphere.2020.128148.
  32. Li, S.Y.; Wu, H.C.; Mai, H.F.; Zhen, J.X.; Li, G.S.; Chen, S.J. Microarray-based analysis of whole-genome DNA methylation profiling in early detection of breast cancer. J Cell Biochem 2019, 120, 658-670, doi:10.1002/jcb.27423.
  33. Dietrich, D.; Lesche, R.; Tetzner, R.; Krispin, M.; Dietrich, J.; Haedicke, W.; Schuster, M.; Kristiansen, G. Analysis of DNA methylation of multiple genes in microdissected cells from formalin-fixed and paraffin-embedded tissues. The journal of histochemistry and cytochemistry : official journal of the Histochemistry Society 2009, 57, 477-489, doi:10.1369/jhc.2009.953026.
  34. Flanagan, J.M.; Brook, M.N.; Orr, N.; Tomczyk, K.; Coulson, P.; Fletcher, O.; Jones, M.E.; Schoemaker, M.J.; Ashworth, A.; Swerdlow, A., et al. Temporal stability and determinants of white blood cell DNA methylation in the breakthrough generations study. Cancer Epidemiol Biomarkers Prev 2015, 24, 221-229, doi:10.1158/1055-9965.Epi-14-0767.
  35. Dumitrescu, R.G. Early Epigenetic Markers for Precision Medicine. Methods in molecular biology (Clifton, N.J.) 2018, 1856, 3-17, doi:10.1007/978-1-4939-8751-1_1.
  36. Dookeran, K.A.; Zhang, W.; Stayner, L.; Argos, M. Associations of two-pore domain potassium channels and triple negative breast cancer subtype in The Cancer Genome Atlas: systematic evaluation of gene expression and methylation. BMC research notes 2017, 10, 475, doi:10.1186/s13104-017-2777-4.
  37. Zhang, S.; Wang, Y.; Gu, Y.; Zhu, J.; Ci, C.; Guo, Z.; Chen, C.; Wei, Y.; Lv, W.; Liu, H., et al. Specific breast cancer prognosis-subtype distinctions based on DNA methylation patterns. Molecular oncology 2018, 12, 1047-1060, doi:10.1002/1878-0261.12309.
  38. Wilson, L.E.; Harlid, S.; Xu, Z.; Sandler, D.P.; Taylor, J.A. An epigenome-wide study of body mass index and DNA methylation in blood using participants from the Sister Study cohort. International journal of obesity (2005) 2017, 41, 194-199, doi:10.1038/ijo.2016.184.
  39. Tessema, M.; Yu, Y.Y.; Stidley, C.A.; Machida, E.O.; Schuebel, K.E.; Baylin, S.B.; Belinsky, S.A. Concomitant promoter methylation of multiple genes in lung adenocarcinomas from current, former and never smokers. Carcinogenesis 2009, 30, 1132-1138, doi:10.1093/carcin/bgp114.
  40. Pangeni, R.P.; Channathodiyil, P.; Huen, D.S.; Eagles, L.W.; Johal, B.K.; Pasha, D.; Hadjistephanou, N.; Nevell, O.; Davies, C.L.; Adewumi, A.I., et al. The GALNT9, BNC1 and CCDC8 genes are frequently epigenetically dysregulated in breast tumours that metastasise to the brain. Clin Epigenetics 2015, 7, 57, doi:10.1186/s13148-015-0089-x.
  41. Klassen, L.M.B.; Chequin, A.; Manica, G.C.M.; Biembengut, I.V.; Toledo, M.B.; Baura, V.A.; de, O.P.F.; Ramos, E.A.S.; Costa, F.F.; de Souza, E.M., et al. MMP9 gene expression regulation by intragenic epigenetic modifications in breast cancer. Gene 2018, 642, 461-466, doi:10.1016/j.gene.2017.11.054.
  42. Kresovich, J.K.; Gann, P.H.; Erdal, S.; Chen, H.Y.; Argos, M.; Rauscher, G.H. Candidate gene DNA methylation associations with breast cancer characteristics and tumor progression. Epigenomics 2018, 10, 367-378, doi:10.2217/epi-2017-0119.
  43. Babicki, S.; Arndt, D.; Marcu, A.; Liang, Y.; Grant, J.R.; Maciejewski, A.; Wishart, D.S. Heatmapper: web-enabled heat mapping for all. Nucleic Acids Res 2016, 44, W147-153, doi:10.1093/nar/gkw419.
  44. Schussel, J.; Zhou, X.C.; Zhang, Z.; Pattani, K.; Bermudez, F.; Jean-Charles, G.; McCaffrey, T.; Padhya, T.; Phelan, J.; Spivakovsky, S., et al. EDNRB and DCC salivary rinse hypermethylation has a similar performance as expert clinical examination in discrimination of oral cancer/dysplasia versus benign lesions. Clin Cancer Res 2013, 19, 3268-3275, doi:10.1158/1078-0432.Ccr-12-3496.
  45. Coleman, W.B.; Rivenbark, A.G. Quantitative DNA methylation analysis: the promise of high-throughput epigenomic diagnostic testing in human neoplastic disease. The Journal of molecular diagnostics : JMD 2006, 8, 152-156, doi:10.2353/jmoldx.2006.060026.
  46. Harden, S.V.; Tokumaru, Y.; Westra, W.H.; Goodman, S.; Ahrendt, S.A.; Yang, S.C.; Sidransky, D. Gene promoter hypermethylation in tumors and lymph nodes of stage I lung cancer patients. Clin Cancer Res 2003, 9, 1370-1375.
  47. Kadam, S.K.; Farmen, M.; Brandt, J.T. Quantitative measurement of cell-free plasma DNA and applications for detecting tumor genetic variation and promoter methylation in a clinical setting. The Journal of molecular diagnostics : JMD 2012, 14, 346-356, doi:10.1016/j.jmoldx.2012.03.001.
  48. Weiss, G.; Schlegel, A.; Kottwitz, D.; König, T.; Tetzner, R. Validation of the SHOX2/PTGER4 DNA Methylation Marker Panel for Plasma-Based Discrimination between Patients with Malignant and Nonmalignant Lung Disease. J Thorac Oncol 2017, 12, 77-84, doi:10.1016/j.jtho.2016.08.123.
  49. Zavridou, M.; Mastoraki, S.; Strati, A.; Tzanikou, E.; Chimonidou, M.; Lianidou, E. Evaluation of Preanalytical Conditions and Implementation of Quality Control Steps for Reliable Gene Expression and DNA Methylation Analyses in Liquid Biopsies. Clin Chem 2018, 64, 1522-1533, doi:10.1373/clinchem.2018.292318.
  50. Alborelli, I.; Generali, D.; Jermann, P.; Cappelletti, M.R.; Ferrero, G.; Scaggiante, B.; Bortul, M.; Zanconati, F.; Nicolet, S.; Haegele, J., et al. Cell-free DNA analysis in healthy individuals by next-generation sequencing: a proof of concept and technical validation study. Cell death & disease 2019, 10, 534, doi:10.1038/s41419-019-1770-3.
  51. Markus, H.; Contente-Cuomo, T.; Farooq, M.; Liang, W.S.; Borad, M.J.; Sivakumar, S.; Gollins, S.; Tran, N.L.; Dhruv, H.D.; Berens, M.E., et al. Evaluation of pre-analytical factors affecting plasma DNA analysis. Scientific reports 2018, 8, 7375, doi:10.1038/s41598-018-25810-0.

Round 2

Reviewer 2 Report

Point 4: In line 300, “After a literature review, aberrant methylated genes that were reported previously were excluded”. Reference is missing and what these genes are not listed anywhere.

Response 4: We thankful the reviewer for the suggestions and comments. Aberrantly methylated genes that had been reported previously were excluded: DPF1 [9], DNM3 [1], TLX1 [10], CHST11 [11], CHST3 [12], ALX1 [6], HNF1B [10], EBF1 [13], PITX2 [14], ITGA5 [15], COL11A2 [2], CFTR [16], NR5A2 [4], PRKCB [4], SOSTDC1 [17], DBX1 [18], LHX8 [19], HCK [20], NXPH1 [21], NID2 [22], TAC1 [23], TFAP2B [24], F2RL3 [25], POU4F2 [26], H2AFY [27], GALR1 [28], FOXD3 [29], NT5E [30], CLIP4 [31], ZNF454 [32], TIMP2 [33], GNG4 [34], MIR129-2 [34], TXNRD1 [35], CPXM1 [34], PRDM14 [34], HOXA4 [36], SLITRK1 [37], AHRR [38], NPTX2 [39], KCNK9 [40], C9orf122 [41], CRHR2 [42], BOLL [43], CCDC8 [44], MMP9 [45], and C12orf68 [41].

Thank you for adding these genes in the manuscript. I think it would be better to give this gene list in a supplemental file. 

Author Response

First revision:

Point 4: In line 300, “After a literature review, aberrant methylated genes that were reported previously were excluded”. Reference is missing and what these genes are not listed anywhere.

Response 4: We thankful the reviewer for the suggestions and comments. Aberrantly methylated genes that had been reported previously were excluded: DPF1 [1], DNM3 [2], TLX1 [3], CHST11 [4], CHST3 [5], ALX1 [6], HNF1B [3], EBF1 [7], PITX2 [8], ITGA5 [9], COL11A2 [10], CFTR [11], NR5A2 [12], PRKCB [12], SOSTDC1 [13], DBX1 [14], LHX8 [15], HCK [16], NXPH1 [17], NID2 [18], TAC1 [19], TFAP2B [20], F2RL3 [21], POU4F2 [22], H2AFY [23], GALR1 [24], FOXD3 [25], NT5E [26], CLIP4 [27], ZNF454 [28], TIMP2 [29], GNG4 [30], MIR129-2 [30], TXNRD1 [31], CPXM1 [30], PRDM14 [30], HOXA4 [32], SLITRK1 [33], AHRR [34], NPTX2 [35], KCNK9 [36], C9orf122 [37], CRHR2 [38], BOLL [39], CCDC8 [40], MMP9 [41], and C12orf68 [37].

Second Revision:

Reviewer’s comments: Thank you for adding these genes in the manuscript. I think it would be better to give this gene list in a supplemental file.

Response to Reviewer’s comments: We thank the reviewer for their suggestions and comments. We have added the gene list in the supplemental file.

Table S3 Gene list for 160 hypermethylated genes was commonly found in Taiwanese and TCGA cohorts

Status

Genes

Aberrant DNA methylated genes in cancers reported previously

AHRR, BARHL2, BOLL, C12orf68, C14orf23, C17orf64, C9orf122, CCDC36, CCDC8, CLIP4, CPXM1, CRHR2, CRYGD, CSDAP1, DPP6, GNG4, HLA-L, HOXA4, HOXD8, ILDR2, KCNK9, MIR129-2, MMP9, MYO15B, NES, NPTX2, NRXN1, NT5E, OLIG3, OTX2OS1, PGLYRP2, PHOX2A, PRDM14, RCN3, SCG5, SCRT2, SEMA6C, SKI, SLITRK1, SOX2OT, SPTBN4, TBR1, TIMP2, TTBK1, TTC28, TULP1, TXNRD1, VWC2, ZNF454, ZNF572 [1-41]

Breast cancer associated genes reported previously

CPEB1, CYTL1, DOCK2, ESRRG, FLI1, FRZB, GRASP, GRIA1, HOXD9, LHX1, MAML3, MEIS2, NCALD, NKX2-1, NPAS4, OCA2, PDX1, RGS17, RGS20, SALL1, SSTR4, TMEM97, TNFAIP8L3, TRABD, VGLL4, WNT3A

Novel aberrant DNA methylated genes

ADCY4, ALX1, BCAT1, C12orf42, C1orf114 (CCDC181), CFTR, CHST11, CHST3, CLDN9, CLEC14A, COL11A2, CRYM, CSMD3, DBX1, DMRTA2, DNM3, DPF1, EBF1, EMX1, ENPP2, EPHX3, EVX2, F2RL3, FAM38B, FOXD3, FSD1, GALR1, GCK, GCM2, GJD2, GRIN1, H2AFY, HCK, HNF1B, HPCAL4, HTR6, IRX1, ITGA5, ITPRIPL1, KCNC3, LHX4, LHX8, LOC643719, LOC646999, MIR663, NID2, NR5A2, NRXN2, NXPH1, OTX1, OTX2, PITX2, POU3F3, POU4F2, PRDM13, PRKAR1B, PRKCB, PRKCE, PRRT1, PTPRN, RNF220, SEZ6L2, SIM1, SLC23A2, SNAP25, SOSTDC1, SRGAP3, SRRM3, SSPO, TAC1, TFAP2B, TLX1, TMEM145, TRH, TRIM46, TRIM71, TRIP10, VANGL2, WIT1, ZIC5, ZNF177, ZNF662, ZSCAN18, TBXT

References

  1. Toth, R.; Scherer, D.; Kelemen, L.E.; Risch, A.; Hazra, A.; Balavarca, Y.; Issa, J.J.; Moreno, V.; Eeles, R.A.; Ogino, S., et al. Genetic Variants in Epigenetic Pathways and Risks of Multiple Cancers in the GAME-ON Consortium. Cancer Epidemiol Biomarkers Prev 2017, 26, 816-825, doi:10.1158/1055-9965.Epi-16-0728.
  2. Uehiro, N.; Sato, F.; Pu, F.; Tanaka, S.; Kawashima, M.; Kawaguchi, K.; Sugimoto, M.; Saji, S.; Toi, M. Circulating cell-free DNA-based epigenetic assay can detect early breast cancer. Breast cancer research : BCR 2016, 18, 129, doi:10.1186/s13058-016-0788-z.
  3. Tommasi, S.; Karm, D.L.; Wu, X.; Yen, Y.; Pfeifer, G.P. Methylation of homeobox genes is a frequent and early epigenetic event in breast cancer. Breast cancer research : BCR 2009, 11, R14, doi:10.1186/bcr2233.
  4. Herman, D.; Leakey, T.I.; Behrens, A.; Yao-Borengasser, A.; Cooney, C.A.; Jousheghany, F.; Phanavanh, B.; Siegel, E.R.; Safar, A.M.; Korourian, S., et al. CHST11 gene expression and DNA methylation in breast cancer. International journal of oncology 2015, 46, 1243-1251, doi:10.3892/ijo.2015.2828.
  5. Nordgard, S.H.; Johansen, F.E.; Alnaes, G.I.; Bucher, E.; Syvänen, A.C.; Naume, B.; Børresen-Dale, A.L.; Kristensen, V.N. Genome-wide analysis identifies 16q deletion associated with survival, molecular subtypes, mRNA expression, and germline haplotypes in breast cancer patients. Genes, chromosomes & cancer 2008, 47, 680-696, doi:10.1002/gcc.20569.
  6. de Groot, J.S.; Moelans, C.B.; Elias, S.G.; Jo Fackler, M.; van Domselaar, R.; Suijkerbuijk, K.P.; Witkamp, A.J.; Sukumar, S.; van Diest, P.J.; van der Wall, E. DNA promoter hypermethylation in nipple fluid: a potential tool for early breast cancer detection. Oncotarget 2016, 7, 24778-24791, doi:10.18632/oncotarget.8352.
  7. Fernandez-Jimenez, N.; Sklias, A.; Ecsedi, S.; Cahais, V.; Degli-Esposti, D.; Jay, A.; Ancey, P.B.; Woo, H.D.; Hernandez-Vargas, H.; Herceg, Z. Lowly methylated region analysis identifies EBF1 as a potential epigenetic modifier in breast cancer. Epigenetics 2017, 12, 964-972, doi:10.1080/15592294.2017.1373919.
  8. Absmaier, M.; Napieralski, R.; Schuster, T.; Aubele, M.; Walch, A.; Magdolen, V.; Dorn, J.; Gross, E.; Harbeck, N.; Noske, A., et al. PITX2 DNA-methylation predicts response to anthracycline-based adjuvant chemotherapy in triple-negative breast cancer patients. International journal of oncology 2018, 52, 755-767, doi:10.3892/ijo.2018.4241.
  9. Fang, Z.; Yao, W.; Xiong, Y.; Zhang, J.; Liu, L.; Li, J.; Zhang, C.; Wan, J. Functional elucidation and methylation-mediated downregulation of ITGA5 gene in breast cancer cell line MDA-MB-468. J Cell Biochem 2010, 110, 1130-1141, doi:10.1002/jcb.22626.
  10. Roos, L.; van Dongen, J.; Bell, C.G.; Burri, A.; Deloukas, P.; Boomsma, D.I.; Spector, T.D.; Bell, J.T. Integrative DNA methylome analysis of pan-cancer biomarkers in cancer discordant monozygotic twin-pairs. Clin Epigenetics 2016, 8, 7, doi:10.1186/s13148-016-0172-y.
  11. Liu, K.; Dong, F.; Gao, H.; Guo, Y.; Li, H.; Yang, F.; Zhao, P.; Dai, Y.; Wang, J.; Zhou, W., et al. Promoter hypermethylation of the CFTR gene as a novel diagnostic and prognostic marker of breast cancer. Cell biology international 2020, 44, 603-609, doi:10.1002/cbin.11260.
  12. Bacolod, M.D.; Huang, J.; Giardina, S.F.; Feinberg, P.B.; Mirza, A.H.; Swistel, A.; Soper, S.A.; Barany, F. Prediction of blood-based biomarkers and subsequent design of bisulfite PCR-LDR-qPCR assay for breast cancer detection. BMC cancer 2020, 20, 85, doi:10.1186/s12885-020-6574-4.
  13. Xiao, B.; Chen, L.; Ke, Y.; Hang, J.; Cao, L.; Zhang, R.; Zhang, W.; Liao, Y.; Gao, Y.; Chen, J., et al. Identification of methylation sites and signature genes with prognostic value for luminal breast cancer. BMC cancer 2018, 18, 405, doi:10.1186/s12885-018-4314-9.
  14. Lindqvist, B.M.; Wingren, S.; Motlagh, P.B.; Nilsson, T.K. Whole genome DNA methylation signature of HER2-positive breast cancer. Epigenetics 2014, 9, 1149-1162, doi:10.4161/epi.29632.
  15. Daino, K.; Nishimura, M.; Imaoka, T.; Takabatake, M.; Morioka, T.; Nishimura, Y.; Shimada, Y.; Kakinuma, S. Epigenetic dysregulation of key developmental genes in radiation-induced rat mammary carcinomas. Int J Cancer 2018, 143, 343-354, doi:10.1002/ijc.31309.
  16. Zhang, C.; Zhao, H.; Li, J.; Liu, H.; Wang, F.; Wei, Y.; Su, J.; Zhang, D.; Liu, T.; Zhang, Y. The identification of specific methylation patterns across different cancers. PloS one 2015, 10, e0120361, doi:10.1371/journal.pone.0120361.
  17. Faryna, M.; Konermann, C.; Aulmann, S.; Bermejo, J.L.; Brugger, M.; Diederichs, S.; Rom, J.; Weichenhan, D.; Claus, R.; Rehli, M., et al. Genome-wide methylation screen in low-grade breast cancer identifies novel epigenetically altered genes as potential biomarkers for tumor diagnosis. FASEB journal : official publication of the Federation of American Societies for Experimental Biology 2012, 26, 4937-4950, doi:10.1096/fj.12-209502.
  18. Strelnikov, V.V.; Kuznetsova, E.B.; Tanas, A.S.; Rudenko, V.V.; Kalinkin, A.I.; Poddubskaya, E.V.; Kekeeva, T.V.; Chesnokova, G.G.; Trotsenko, I.D.; Larin, S.S., et al. Abnormal promoter DNA hypermethylation of the integrin, nidogen, and dystroglycan genes in breast cancer. Scientific reports 2021, 11, 2264, doi:10.1038/s41598-021-81851-y.
  19. Jeschke, J.; Van Neste, L.; Glöckner, S.C.; Dhir, M.; Calmon, M.F.; Deregowski, V.; Van Criekinge, W.; Vlassenbroeck, I.; Koch, A.; Chan, T.A., et al. Biomarkers for detection and prognosis of breast cancer identified by a functional hypermethylome screen. Epigenetics 2012, 7, 701-709, doi:10.4161/epi.20445.
  20. Ennour-Idrissi, K.; Dragic, D.; Issa, E.; Michaud, A.; Chang, S.L.; Provencher, L.; Durocher, F.; Diorio, C. DNA Methylation and Breast Cancer Risk: An Epigenome-Wide Study of Normal Breast Tissue and Blood. Cancers 2020, 12, doi:10.3390/cancers12113088.
  21. Zhang, Y.; Yang, R.; Burwinkel, B.; Breitling, L.P.; Holleczek, B.; Schöttker, B.; Brenner, H. F2RL3 methylation in blood DNA is a strong predictor of mortality. International journal of epidemiology 2014, 43, 1215-1225, doi:10.1093/ije/dyu006.
  22. Lee, S.E.; Kim, S.J.; Yoon, H.J.; Yu, S.Y.; Yang, H.; Jeong, S.I.; Hwang, S.Y.; Park, C.S.; Park, Y.S. Genome-wide profiling in melatonin-exposed human breast cancer cell lines identifies differentially methylated genes involved in the anticancer effect of melatonin. Journal of pineal research 2013, 54, 80-88, doi:10.1111/j.1600-079X.2012.01027.x.
  23. Halvorsen, A.R.; Helland, A.; Fleischer, T.; Haug, K.M.; Grenaker Alnaes, G.I.; Nebdal, D.; Syljuåsen, R.G.; Touleimat, N.; Busato, F.; Tost, J., et al. Differential DNA methylation analysis of breast cancer reveals the impact of immune signaling in radiation therapy. Int J Cancer 2014, 135, 2085-2095, doi:10.1002/ijc.28862.
  24. Kanazawa, T.; Misawa, K.; Fukushima, H.; Misawa, Y.; Sato, Y.; Maruta, M.; Imayoshi, S.; Kusaka, G.; Kawabata, K.; Mineta, H., et al. Epigenetic inactivation of galanin receptors in salivary duct carcinoma of the parotid gland: Potential utility as biomarkers for prognosis. Oncology letters 2018, 15, 9043-9050, doi:10.3892/ol.2018.8525.
  25. Khakpour, G.; Noruzinia, M.; Izadi, P.; Karami, F.; Ahmadvand, M.; Heshmat, R.; Amoli, M.M.; Tavakkoly-Bazzaz, J. Methylomics of breast cancer: Seeking epimarkers in peripheral blood of young subjects. Tumour Biol 2017, 39, 1010428317695040, doi:10.1177/1010428317695040.
  26. Lo Nigro, C.; Monteverde, M.; Lee, S.; Lattanzio, L.; Vivenza, D.; Comino, A.; Syed, N.; McHugh, A.; Wang, H.; Proby, C., et al. NT5E CpG island methylation is a favourable breast cancer biomarker. Br J Cancer 2012, 107, 75-83, doi:10.1038/bjc.2012.212.
  27. Fan, Y.; He, L.; Wang, Y.; Fu, S.; Han, Y.; Fan, J.; Wen, Q. CLIP4 Shows Putative Tumor Suppressor Characteristics in Breast Cancer: An Integrated Analysis. Front Mol Biosci 2020, 7, 616190, doi:10.3389/fmolb.2020.616190.
  28. Zhu, Q.; Wang, J.; Zhang, Q.; Wang, F.; Fang, L.; Song, B.; Xie, C.; Liu, J. Methylationdriven genes PMPCAP1, SOWAHC and ZNF454 as potential prognostic biomarkers in lung squamous cell carcinoma. Mol Med Rep 2020, 21, 1285-1295, doi:10.3892/mmr.2020.10933.
  29. Simonova, O.A.; Kuznetsova, E.B.; Tanas, A.S.; Rudenko, V.V.; Poddubskaya, E.V.; Kekeeva, T.V.; Trotsenko, I.D.; Larin, S.S.; Kutsev, S.I.; Zaletaev, D.V., et al. Abnormal Hypermethylation of CpG Dinucleotides in Promoter Regions of Matrix Metalloproteinases Genes in Breast Cancer and Its Relation to Epigenomic Subtypes and HER2 Overexpression. Biomedicines 2020, 8, doi:10.3390/biomedicines8050116.
  30. Mao, X.H.; Ye, Q.; Zhang, G.B.; Jiang, J.Y.; Zhao, H.Y.; Shao, Y.F.; Ye, Z.Q.; Xuan, Z.X.; Huang, P. Identification of differentially methylated genes as diagnostic and prognostic biomarkers of breast cancer. World J Surg Oncol 2021, 19, 29, doi:10.1186/s12957-021-02124-6.
  31. Liang, Z.Z.; Zhu, R.M.; Li, Y.L.; Jiang, H.M.; Li, R.B.; Tang, L.Y.; Wang, Q.; Ren, Z.F. Differential epigenetic and transcriptional profile in MCF-7 breast cancer cells exposed to cadmium. Chemosphere 2020, 261, 128148, doi:10.1016/j.chemosphere.2020.128148.
  32. Li, S.Y.; Wu, H.C.; Mai, H.F.; Zhen, J.X.; Li, G.S.; Chen, S.J. Microarray-based analysis of whole-genome DNA methylation profiling in early detection of breast cancer. J Cell Biochem 2019, 120, 658-670, doi:10.1002/jcb.27423.
  33. Dietrich, D.; Lesche, R.; Tetzner, R.; Krispin, M.; Dietrich, J.; Haedicke, W.; Schuster, M.; Kristiansen, G. Analysis of DNA methylation of multiple genes in microdissected cells from formalin-fixed and paraffin-embedded tissues. The journal of histochemistry and cytochemistry : official journal of the Histochemistry Society 2009, 57, 477-489, doi:10.1369/jhc.2009.953026.
  34. Flanagan, J.M.; Brook, M.N.; Orr, N.; Tomczyk, K.; Coulson, P.; Fletcher, O.; Jones, M.E.; Schoemaker, M.J.; Ashworth, A.; Swerdlow, A., et al. Temporal stability and determinants of white blood cell DNA methylation in the breakthrough generations study. Cancer Epidemiol Biomarkers Prev 2015, 24, 221-229, doi:10.1158/1055-9965.Epi-14-0767.
  35. Dumitrescu, R.G. Early Epigenetic Markers for Precision Medicine. Methods in molecular biology (Clifton, N.J.) 2018, 1856, 3-17, doi:10.1007/978-1-4939-8751-1_1.
  36. Dookeran, K.A.; Zhang, W.; Stayner, L.; Argos, M. Associations of two-pore domain potassium channels and triple negative breast cancer subtype in The Cancer Genome Atlas: systematic evaluation of gene expression and methylation. BMC research notes 2017, 10, 475, doi:10.1186/s13104-017-2777-4.
  37. Zhang, S.; Wang, Y.; Gu, Y.; Zhu, J.; Ci, C.; Guo, Z.; Chen, C.; Wei, Y.; Lv, W.; Liu, H., et al. Specific breast cancer prognosis-subtype distinctions based on DNA methylation patterns. Molecular oncology 2018, 12, 1047-1060, doi:10.1002/1878-0261.12309.
  38. Wilson, L.E.; Harlid, S.; Xu, Z.; Sandler, D.P.; Taylor, J.A. An epigenome-wide study of body mass index and DNA methylation in blood using participants from the Sister Study cohort. International journal of obesity (2005) 2017, 41, 194-199, doi:10.1038/ijo.2016.184.
  39. Tessema, M.; Yu, Y.Y.; Stidley, C.A.; Machida, E.O.; Schuebel, K.E.; Baylin, S.B.; Belinsky, S.A. Concomitant promoter methylation of multiple genes in lung adenocarcinomas from current, former and never smokers. Carcinogenesis 2009, 30, 1132-1138, doi:10.1093/carcin/bgp114.
  40. Pangeni, R.P.; Channathodiyil, P.; Huen, D.S.; Eagles, L.W.; Johal, B.K.; Pasha, D.; Hadjistephanou, N.; Nevell, O.; Davies, C.L.; Adewumi, A.I., et al. The GALNT9, BNC1 and CCDC8 genes are frequently epigenetically dysregulated in breast tumours that metastasise to the brain. Clin Epigenetics 2015, 7, 57, doi:10.1186/s13148-015-0089-x.
  41. Klassen, L.M.B.; Chequin, A.; Manica, G.C.M.; Biembengut, I.V.; Toledo, M.B.; Baura, V.A.; de, O.P.F.; Ramos, E.A.S.; Costa, F.F.; de Souza, E.M., et al. MMP9 gene expression regulation by intragenic epigenetic modifications in breast cancer. Gene 2018, 642, 461-466, doi:10.1016/j.gene.2017.11.054.

Reviewer 3 Report

Thanks for the authors' reply.

For Figure S2, the Y-axis do not change. Please double check.

I strongly suggest to at least plot the absolute DNA methylation levels (0-1 or 0%-100%) for all the genes in this manuscript in the supplement. The reason is that the absolute DNA methylation values is the golden standard, which will be easier and valuable for the future studies to refer or compare with the results in this study. qMSP cannot directly compare among different labs.

It is better to label "qMSP" on the Y-axis.

Author Response

Response to Reviewer 3 comments:

Point: For Figure S2, the Y-axis do not change. Please double check.

I strongly suggest to at least plot the absolute DNA methylation levels (0-1 or 0%-100%) for all the genes in this manuscript in the supplement. The reason is that the absolute DNA methylation values is the golden standard, which will be easier and valuable for the future studies to refer or compare with the results in this study. qMSP cannot directly compare among different labs.

It is better to label "qMSP" on the Y-axis.

Response:

We apologize for the labeling error on the Y-axis. We thank the reviewer for the suggestions and comments. We have corrected the error in Table S2. We labels as ”The relative DNA methylation with qMSP normalized by ACTB gene”.

Indeed, relative DNA methylation assays are not a good replacement for absolute assays. However, carefully selected, designed and validated qMSP can cost-effectively detect minimal traces of methylated DNA against an excess of unmethylated DNA [1], especially in hundreds of limited clinical tissue samples and plasma circulating cell free DNA from patients with early stage breast cancer. We have added the description of study limitation to lines 700-706 of the discussion section as follow: In addition, the fact that relative DNA methylation assay by qMSP was used to analyze the DNA methylation levels rather than absolute DNA methylation assays is the limitation of this study. Relative DNA methylation assay by qMSP is not a good replacement for absolute assays. However, carefully selected, designed and validated qMSP assays can cost-effectively detect trace levels of methylated DNA against an excess of unmethylated DNA [1], especially in hundreds of limited clinical tissue samples or plasma circulating cell free DNA from patients with early stage breast cancer.

References

  1. Bock C., H.F., Carmona F.J., Tierling S., Datlinger P., Assenov Y., Berdasco M., Bergmann A.K., Booher K., Busato F., et al. Quantitative comparison of DNA methylation assays for biomarker development and clinical applications. Nature biotechnology 2016, 34, 726-737, doi:10.1038/nbt.3605.
